# Layer-Adaptive State Pruning
# for Deep State Space Models

**Minseon Gwak†, Seongrok Moon†, Joohwan Ko‡, PooGyeon Park†\***
† Department of Electrical Engineering, POSTECH
‡ Department of Computer Science, University of Massachusetts Amherst
{minseon25,srmoon,ppg}@postech.ac.kr, joohwanko@cs.umass.edu

## Abstract

Due to the lack of state dimension optimization methods, deep state space models (SSMs) have sacrificed model capacity, training search space, or stability to alleviate computational costs caused by high state dimensions. In this work, we provide a structured pruning method for SSMs, **L**ayer-**A**daptive **ST**ate pruning (LAST), which reduces the state dimension of each layer in minimizing model-level output energy loss by extending modal truncation for a single system. LAST scores are evaluated using the $\mathcal{H}_\infty$ norms of subsystems and layer-wise energy normalization. The scores serve as global pruning criteria, enabling cross-layer comparison of states and layer-adaptive pruning. Across various sequence benchmarks, LAST optimizes previous SSMs, revealing the redundancy and compressibility of their state spaces. Notably, we demonstrate that, on average, pruning $33\%$ of states still maintains performance with $0.52\%$ accuracy loss in multi-input multi-output SSMs without retraining. Code is available at https://github.com/msgwak/LAST.

## 1 Introduction

Deep state space models (SSMs) have proven effective in modeling sequential data by optimally compressing input history to internal states [Gu et al., 2020, 2021, 2022b, Gu and Dao, 2023, Zhang et al., 2023, Parnichkun et al., 2024]. Given their modeling capabilities, ensuring the feasibility and stability of SSMs during training has become a crucial research focus for achieving efficient learning without divergence. Leveraging the knowledge founded in linear system theory [Kailath, 1980], various advancements have emerged, including stability-guaranteeing parameterization [Gu et al., 2022a], general system architecture [Smith et al., 2023], and efficiency improvements via frequency-domain operations, utilizing the fast Fourier transform and the transfer functions of systems [Gu et al., 2022b,a, Zhang et al., 2023, Parnichkun et al., 2024].

One of the main computation and memory contributors of SSMs is the state dimension $n$. Since the initial proposal of SSMs, a multiple single-input single-output (multi-SISO) architecture has been employed for scalable and efficient training Gu et al. [2022b,a], Gu and Dao [2023], Zhang et al. [2023], Parnichkun et al. [2024]. In this architecture, rather than directly learning an $n$-dimensional system, smaller-dimensional SISO systems are trained in parallel and then integrated through a channel-mixing layer. Within this structure, Gupta et al. [2022] presented that diagonal systems can achieve matching performance to nondiagonal systems. Gu et al. [2022a] introduced a stability-guaranteed model, where the diagonal systems are trained to satisfy the necessary and sufficient stability condition.

Instead of utilizing multiple SISO systems in parallel, Smith et al. [2023] adopted a multi-input multi-output (MIMO) architecture, where the enhanced information usage through a MIMO system.

---

*corresponding author

38th Conference on Neural Information Processing Systems (NeurIPS 2024).

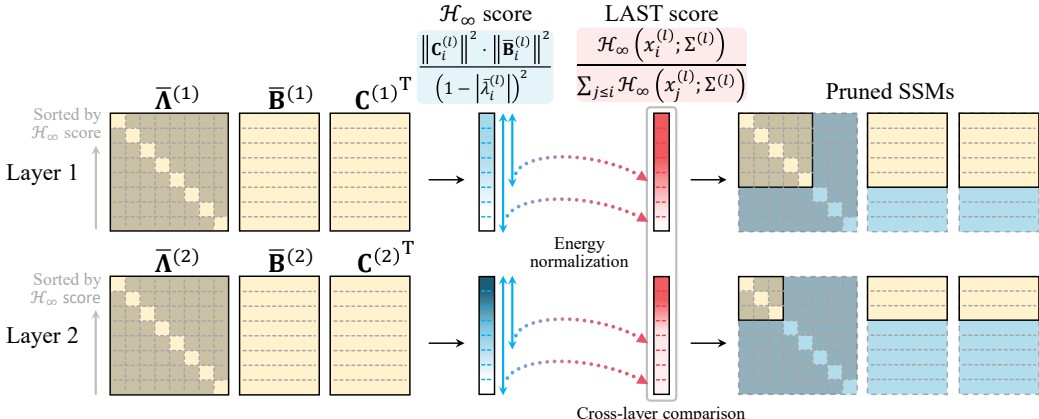

Figure 1: Illustration of LAST for two layers. Matrices are divided by lines on a per-state basis, and subsystems are sorted in descending order by their $\mathcal{H}_\infty$ norms. LAST scores are obtained by normalizing each $\mathcal{H}_\infty$ norm by the sum of all $\mathcal{H}_\infty$ norms in a layer when the states with lower $\mathcal{H}_\infty$ norms are excluded. Since LAST scores correlate with model-level output energy loss, we prune all parameters corresponding to states with low LAST scores.

This architecture provides high performance with much smaller state dimensions than equivalent block systems in multi-SISO layers. For instance, in the `Path-X` task that involves the longest tested sequences, this architecture showed state-of-the-art performance [Smith et al., 2023, Parnichkun et al., 2024]. However, both architectures lack optimization methods for state dimensions, leading to inefficiencies when the model is over-parameterized for the task.

Recently, Parnichkun et al. [2024] parameterized the transfer functions of SISO systems and proposed a state-free inference. However, this approach indirectly trains the poles of the transfer functions, resulting in a restrictive search space or stability being guaranteed only at initialization.

Focusing on the stability-guaranteed diagonal SSMs, we develop and verify a layer-adaptive model order reduction (MOR) method for SSMs to identify the least significant states or subsystems in terms of their impact on task performance. Inspired by layer-adaptive neural network pruning [Evci et al., 2020, Lee et al., 2021, Xu et al., 2023] and extending the traditional MOR for a single system [Green and Limebeer, 2012], we propose **L**ayer-**A**daptive **ST**ate pruning (LAST), where importance scores for learned states are evaluated and used as global pruning criteria. LAST scores measure the relative maximum frequency-domain gain of each subsystem when subsystems with lower scores are excluded, as illustrated in Figure 1. LAST prunes insignificant subsystems to achieve a desired compression level, reducing unnecessary computational and memory costs while bounding the output distortion by the $\mathcal{H}_\infty$ norms of the pruned subsystems.

We validate the insignificant state identification performance of LAST on long-range sequences, including Long Range Arena (LRA) [Tay et al., 2021] and Speech Command [Warden, 2018] benchmarks. Our results present that previous SSMs have great *compressibility*, demonstrating that pruning 33% (26.25%) of the trained states resulted in only 0.52% (0.32%) of accuracy loss in MIMO models (in multi-SISO models) on average, including the non-compressible cases.

## 2 Background

### 2.1 Stability of state space models

A DT SSM is stable if all poles, roots of a denominator, of its transfer function lie within the unit circle. However, it is challenging to train systems to ensure stability at every step. One approach for this issue is to confine the search space to sufficient stable region [Zhang et al., 2023], as illustrated in Figure 5 for a second-order linear time-invariant (LTI) system. Due to the restricted search space, training under this condition can limit model performance [Parnichkun et al., 2024]. Another approach is initializing the system at the center of the stable region, as marked in Figure 5, referred

to as zero initialization in [Parnichkun et al., 2024]. While this approach mitigates the performance limitation, the stability is guaranteed only at initialization.

In contrast, diagonal SSMs [Gupta et al., 2022, Gu et al., 2022a, Smith et al., 2023] directly parameterize the system poles, enabling the model to explore all expressible systems that possess stability-satisfying poles. Thus, all our derivations are based on the diagonal SSMs to leverage the guaranteed stability, which allows for the application of various system analysis techniques. Detailed explanations on the stability regions are provided in Appendix A.1.

## 2.2 Diagonal state space models

**Architectures.** Diagonal SSMs consist of an encoder that increases the number of input channels to $h$, $L$ SSM layers, and a decoder for the downstream task. Each SSM layer can be designed with either a multi-SISO or MIMO architecture. In the multi-SISO architecture [Gu et al., 2022a], independent systems are trained for each input channel, with a total of $h$ $n_s$th-order SISO systems being learned in a layer. A fully connected layer is then used to mix features from different channels. In contrast, the MIMO architecture [Smith et al., 2023] employs an $n_m$th-order MIMO system within each layer, handling $h$-dimensional input and output signals. As noted in Smith et al. [2023], $h$ SISO systems in a layer can be represented as one MIMO system, where specific states are assigned to each input channel. Therefore, we describe SSM layers using MIMO expressions, defining the effective total state dimension for an SSM layer by $n$, where $n = n_s h$ for a multi-SISO layer and $n = n_m$ for a MIMO layer.

**Parameterization.** The learnable parameters in a diagonal SSM layer with state dimension $n$ are CT system matrices $\mathbf{\Lambda} \in \mathbb{C}^{n \times n}$, $\mathbf{B} \in \mathbb{C}^{n \times h}$, $\mathbf{C} \in \mathbb{C}^{h \times n}$, $\mathbf{D} \in \mathbb{R}^{h \times h}$, where $\mathbf{\Lambda} \in \mathbb{C}^{n \times n}$ is a diagonal matrix and complex-valued matrices consist of elements that form conjugate pairs to handle real-valued signals [Gu et al., 2022a]. In the diagonal structure, each subsystem is discretized by applying different timescales from $\mathbf{\Delta} \in \mathbb{R}^n$ to process discrete sequences. By zero-order hold (ZOH) discretization [Chen, 1984], a discretized LTI diagonal system $\Sigma : \mathbf{u} \mapsto \mathbf{y}$ in a layer $f_\sigma(\mathbf{u}_k; \Sigma)$ can be represented as follows:

$$\mathbf{x}_{k+1} = \overline{\mathbf{\Lambda}} \mathbf{x}_k + \overline{\mathbf{B}} \mathbf{u}_k, \qquad \mathbf{y}_k = \mathbf{C} \mathbf{x}_k + \mathbf{D} \mathbf{u}_k, \tag{1}$$

where $\overline{\mathbf{\Lambda}} = e^{\mathbf{\Lambda} \mathbf{\Delta}}$ and $\overline{\mathbf{B}} = \mathbf{\Lambda}^{-1}(\overline{\mathbf{\Lambda}} - \mathbf{I}_n)\mathbf{B}$ are the discretized system matrices, $\mathbf{x}_k \in \mathbb{C}^n$ is a state vector, $\mathbf{u}_k \in \mathbb{R}^h$ is an input signal, and $\mathbf{y}_k \in \mathbb{R}^h$ is an output signal. The stability of the discretized system can be achieved by ensuring the stability of the CT parameters with Hurwitz parameterization [Gu et al., 2022a], as derived in Appendix A.2. Finally, a nonlinear activation function $\sigma(\cdot)$ is applied to the output of the linear system, i.e., $f_\sigma(\mathbf{u}_k; \Sigma) = \sigma(\Sigma(\mathbf{u}_k))$, introducing nonlinearity to the SSM.

## 2.3 $\mathcal{H}_\infty$ norms of systems

In robust control, the $\mathcal{H}_\infty$ norm is widely used to minimize the worst-case gain from disturbances to outputs, ensuring stability and performance under system uncertainty [Qin and Sun, 2023, Zheng et al., 2023]. In this work, we use the $\mathcal{H}_\infty$ norm to measure the divergence between the original and approximated systems.

For a DT LTI system $\Sigma : \mathbf{u} \mapsto \mathbf{y}$ with the transfer function matrix (TFM) $\mathbf{G}$, the $\mathcal{H}_\infty$ norm of the system is defined by

$$\|\mathbf{G}\|_\infty := \sup_{\theta \in [0, 2\pi]} \overline{\sigma}(\mathbf{G}(e^{j\theta})),$$

where $\overline{\sigma}$ denotes the maximum singular value of a matrix. In robust control design, the $\mathcal{H}_\infty$ norm is frequently minimized to design controllers that ensure the system performs optimally under disturbance.

In this work, we utilize the following important property of the $\mathcal{H}_\infty$ norm, that is, the energy of the output signal $\|\mathbf{y}\|_2^2$ can be bounded with the squared $\mathcal{H}_\infty$ norm and the energy of the input signal $\|\mathbf{u}\|_2^2$, i.e.,

$$\|\mathbf{y}\|_2^2 \le \|\mathbf{G}\|_\infty^2 \|\mathbf{u}\|_2^2, \tag{2}$$

(See Appendix B.1 for derivation). In other words, the $\mathcal{H}_\infty$ norm of a system measures the maximum gain of the system, which is useful in assessing the energy loss caused by pruning.

# 3 LAST: Layer-adaptive state pruning for SSMs

We propose a structured pruning for SSMs with per-state pruning granularity, where all parameters associated with the identified insignificant state are pruned. Although pruning is implemented by *masking*, we represent pruned systems with their effective remaining states and parameters.

In Section 3.1, we derive a local pruning criterion by evaluating the layer-level energy loss for a single SSM layer, which consists of a MIMO system followed by nonlinear activation. In Section 3.2, we extend this to a global pruning criterion by assessing the model-level energy loss when considering multiple stacked SSM layers.

## 3.1 $\mathcal{H}_\infty$ scores as local pruning criteria

From a DT system $\Sigma : (\overline{\mathbf{\Lambda}}, \overline{\mathbf{B}}, \mathbf{C})$, suppose we prune the $i$th subsystem $\Sigma_i : (\overline{\mathbf{\Lambda}}_i, \overline{\mathbf{B}}_i, \mathbf{C}_i)$ corresponding to the $i$th state $\mathbf{x}_i$, leaving the $i$th state-pruned system $\Sigma_{-i}$. Specifically, the state-pruned system can be written as follows:

$$\Sigma_{-i} : \left( \begin{array}{l} \overline{\mathbf{\Lambda}}_{-i} = \mathrm{diag}(\overline{\lambda}_1, \cdots, \overline{\lambda}_{i-1}, \overline{\lambda}_{i+1}, \cdots, \overline{\lambda}_n), \\ \overline{\mathbf{B}}_{-i}^\top = [\ \overline{\mathbf{B}}_1^\top \quad \cdots \quad \overline{\mathbf{B}}_{i-1}^\top \quad \overline{\mathbf{B}}_{i+1}^\top \quad \cdots \quad \overline{\mathbf{B}}_n^\top \ ], \\ \mathbf{C}_{-i} = [\ \mathbf{C}_1 \quad \cdots \quad \mathbf{C}_{i-1} \quad \mathbf{C}_{i+1} \quad \cdots \quad \mathbf{C}_n \ ] \end{array} \right),$$

where $\overline{\mathbf{\Lambda}} = \mathrm{diag}(\overline{\lambda}_1, \cdots, \overline{\lambda}_n), \overline{\mathbf{B}}^\top = [\ \overline{\mathbf{B}}_1^\top \quad \cdots \quad \overline{\mathbf{B}}_n^\top \ ]$, with $\mathbf{C} = [\ \mathbf{C}_1 \quad \cdots \quad \mathbf{C}_n \ ]$ for $\overline{\mathbf{B}}_i \in \mathbb{C}^{1 \times h}$ and $\mathbb{C}^{h \times 1}$. Our objective is to minimize the layer-level output energy loss, defined as the squared $\ell_2$ distortion in the output signal, incurred by the system approximation through state pruning. The optimization is formalized by

$$\underset{\mathcal{P} \subset \mathcal{S}}{\mathrm{minimize}} \quad \|f_\sigma(\mathbf{u}; \Sigma) - f_\sigma(\mathbf{u}; \Sigma_{-\mathcal{P}})\|_2^2 \tag{3}$$
$$\text{subject to} \quad |\mathcal{P}| \geq r,$$

where $\mathcal{S} = \{1, \cdots, n\}$ is the set of state indices in the full system, $\mathcal{P}$ is the set of pruned state indices, and $r$ indicates the required level of model reduction.

Using the properties of diagonal systems and the $\mathcal{H}_\infty$ norm in Equation (2), the energy loss can be bounded as follows:

$$\|f_\sigma(\mathbf{u}; \Sigma) - f_\sigma(\mathbf{u}; \Sigma_{-\mathcal{P}})\|_2^2 \leq \sum_{i \in \mathcal{P}} \|\mathbf{G}_i\|_\infty^2 \|\mathbf{u}\|_2^2, \tag{4}$$

where $-\mathcal{P} := \mathcal{S} \setminus \mathcal{P}$ and $\mathbf{G}_i$ is the TFM of $\Sigma_i$ (See Appendix B.2 for proof). Therefore, we can reduce a system by pruning subsystems with small $\mathcal{H}_\infty$ norms, minimizing the upper bound in Equation (3). This result shows that, even in the presence of nonlinearity, pruning for a single layer can be performed similarly to modal truncation [Green and Limebeer, 2012]. As the stability is guaranteed by Hurwitz parameterization [Gu et al., 2022a], the $\mathcal{H}_\infty$ norm of a subsystem is evaluated as follows:

$$\|\mathbf{G}_i\|_\infty = \frac{\|\mathbf{C}_i \overline{\mathbf{B}}_i\|}{1 - |\overline{\lambda}_i|}. \tag{5}$$

Hence, the importance of $\mathbf{x}_i$ can be defined by the squared $\mathcal{H}_\infty$ norm of $\Sigma_i$ with a minor optimization of computational efficiency for rank-1 matrix $\mathbf{C}_i \overline{\mathbf{B}}_i$ as follows:

$$\mathcal{H}_\infty(\mathbf{x}_i; \Sigma) = \frac{\|\mathbf{C}_i\|^2 \|\overline{\mathbf{B}}_i\|^2}{\left(1 - |\overline{\lambda}_i|\right)^2}, \tag{6}$$

where $\mathcal{H}_\infty(\mathbf{x}_i; \Sigma)$ refers to the $\mathcal{H}_\infty$ score of $\mathbf{x}_i$, and we prioritize pruning states with lower scores. This can also be simplified with $\|\overline{\mathbf{B}}_i\|^2 = 1$ when $\overline{\mathbf{B}}$ is fixed while $\mathbf{C}$ is trained. Moreover, when two $\mathbf{C}$ matrices are used for bidirectional SSMs, $\|\mathbf{C}_i\|^2$ can be substituted as the average for the two matrices, i.e., $\|\mathbf{C}_i\|^2 = (\|\mathbf{C}_i^f\|^2 + \|\mathbf{C}_i^b\|^2)/2$, where $\|\mathbf{C}_i^f\|$ is for forward direction and $\|\mathbf{C}_i^b\|$ is for backward direction.

The $\mathcal{H}_\infty$ score can be used as a local pruning criterion once the target pruning ratio for each layer is determined. However, this approach has limitations, as it requires a heuristic to determine the pruning ratio for each layer and applies the same amount of pruning without considering layer-specific characteristics.

## 3.2 LAST scores as global pruning criteria

To extend the local pruning criterion to a global pruning criterion, we now consider the *model-level* output energy loss incurred by pruning $L$ layers. In the following description, superscripts indicate layer indices from 1 to $L$ for signals, systems, and state index sets.

Following the notation in Lee et al. [2021], Xu et al. [2023], the output of $k$th layer is obtained by recursively applying the preceding systems and activation functions as follows:

$$f_\sigma(\mathbf{u}^{(1)}; \Sigma^{(1:k)}) = \sigma(\Sigma^{(k)}(f_\sigma(\mathbf{u}^{(1)}; \Sigma^{(1:k-1)}))).$$

Our objective is to minimize the model-level output energy loss as follows:

$$\underset{\mathcal{P}^{(l)} \subset \mathcal{S}^{(l)}}{\text{minimize}} \quad \|f_\sigma(\mathbf{u}^{(1)}; \Sigma^{(1:L)}) - f_\sigma(\mathbf{u}^{(1)}; \widehat{\Sigma}^{(1:L)})\|_2^2$$

$$\text{subject to} \quad \sum_{l=1}^{L} |\mathcal{P}^{(l)}| \geq R,$$

where $\widehat{\Sigma}^{(l)} := \Sigma_{-\mathcal{P}^{(l)}}^{(l)}$ and $R$ represents the total required level of model reduction across all layers. Similar to Lee et al. [2021], we consider a greedy iterative optimization, where we decide the next pruning state $x_i^{(l)}$ by optimizing the following problem for every step:

$$\underset{l \in \{1, \cdots, L\}, \ i \in \mathcal{S}_t^{(l)}}{\text{minimize}} \quad J_l\big(i; \widetilde{\Sigma}_t^{(1:L)}\big), \tag{7}$$

where $J_l\big(i; \widetilde{\Sigma}_t^{(1:L)}\big) := \big\|f_\sigma(\mathbf{u}^{(1)}; \widetilde{\Sigma}_t^{(1:L)}) - f_\sigma(\mathbf{u}^{(1)}; \widetilde{\Sigma}_t^{(1:l-1)}, \Sigma_{\mathcal{S}_t^{(l)} \setminus \{i\}}^{(l)}, \widetilde{\Sigma}_t^{(l+1:L)})\big\|_2^2$, $t$ denotes the step index, and $\widetilde{\Sigma}_t^{(l)} := \Sigma_{\mathcal{S}_t^{(l)}}^{(l)}$ with $\mathcal{S}_t^{(l)} \subset \mathcal{S}^{(l)}$ indicating the set of remaining states at $t$ step. The objective function in Equation (7) represents the model-level output energy loss when pruning a single subsystem in one layer among layers pruned to different extents. By the proof provided in the Appendix B.3, the objective function can be upper-bounded by

$$J_l\big(i; \widetilde{\Sigma}_t^{(1:L)}\big) \leq \frac{\big\|\mathbf{G}_i^{(l)}\big\|_\infty^2}{\big\|\mathbf{G}_{\mathcal{S}_t^{(l)}}^{(l)}\big\|_\infty^2} \prod_{k=1}^{L} \big\|\mathbf{G}_{\mathcal{S}_t^{(k)}}^{(k)}\big\|_\infty^2 \big\|\mathbf{u}^{(1)}\big\|_2^2. \tag{8}$$

Therefore, the upper bound for a subsystem correlates with the ratio of the squared $\mathcal{H}_\infty$ norm of the subsystem to the squared $\mathcal{H}_\infty$ norm of the remaining system for layer $l$. The other terms, except for the ratio, in Equation (8) are common across all layers and can, therefore, be excluded from the cross-layer importance score calculation.

Although we initially considered an iterative optimization to determine the next pruning state, the important scores for all states in all layers can be computed with a few steps, as each score is independently determined based on the trained parameters. For efficient evaluation of the scores, we sort the subsystems in each layer in descending order with their $\mathcal{H}_\infty$ norms in advance, s.t., $\mathcal{H}_\infty\big(\mathbf{x}_i^{(l)}; \Sigma^{(l)}\big) > \mathcal{H}_\infty\big(\mathbf{x}_j^{(l)}; \Sigma^{(l)}\big)$ for $i < j$. Finally, we define the LAST score for $\mathbf{x}_i^{(l)}$ as follows:

$$\mathsf{LAST}\big(\mathbf{x}_i^{(l)}; \Sigma^{(l)}\big) = \frac{\mathcal{H}_\infty\big(\mathbf{x}_i^{(l)}; \Sigma^{(l)}\big)}{\sum_{j \leq i} \mathcal{H}_\infty\big(\mathbf{x}_j^{(l)}; \Sigma^{(l)}\big)} \tag{9}$$

$$= \left(\frac{\|\mathbf{C}_i^{(l)}\|^2 \|\overline{\mathbf{B}}_i^{(l)}\|^2}{\big(1 - |\overline{\lambda}_i^{(l)}|\big)^2}\right) \bigg/ \sum_{j \leq i} \left(\frac{\|\mathbf{C}_j^{(l)}\|^2 \|\overline{\mathbf{B}}_j^{(l)}\|^2}{\big(1 - |\overline{\lambda}_j^{(l)}|\big)^2}\right). \tag{10}$$

Similar to the local pruning criterion, Equation (10) can be modified for the case of using fixed $\overline{\mathbf{B}}^{(l)}$ or bidirectional SSMs. The LAST score for each state reveals the contribution of the subsystem to the model output by assessing the relative gain within the remaining system in a layer, thereby indicating the significance of the subsystem. We refer to this relative metric calculation as *energy normalization*, which adjusts the state importance from different layers to a comparable scale, enabling a *cross-layer* comparison of the states from different layers. In this way, states with lower LAST scores are selected from the overall states, and layer-adaptive pruning is performed according to the desired model-level compression rate.

# 4 Related works

**Model order reduction.** In linear system theory, MOR methods have been extensively researched to approximate high-dimensional systems in engineering applications, such as VLSI [Antoulas and Sorensen, 2001], power systems [Li and White, 1999], and various systems that employ spatial discretization [Jones and Kerrigan, 2010, Curtain and Zwart, 2012, Penzl, 2006]. Using the $\mathcal{H}_\infty$ norm to characterize a stable system, modal truncation [Green and Limebeer, 2012] removes states from a diagonal realization for minimal $\mathcal{H}_\infty$ norm distortion of the system. Balanced truncation [Khalil et al., 1996, Safonov and Chiang, 1988] transforms a given system into a form, not necessarily diagonal, where all states are controllable and observable, then truncates the transformed system. Due to its superior approximation quality, balanced truncation has been developed for various systems and conditions [Petreczky et al., 2013, Besselink et al., 2014, Cheng et al., 2019]. However, transformation into non-diagonal systems is not applicable to current diagonal SSMs, where diagonal parameterization is necessary for computational efficiency and stability. In our work, per-state pruning granularity operates similarly to modal truncation. Compared to traditional MOR, which is reducing a single linear system, we extend it to multi-system reduction, where nonlinear functions are also involved, which have not been addressed in traditional system theory.

**Layer-adaptive neural network pruning.** Using magnitude as a pruning criterion, previous works have demonstrated the superiority of layer-adaptive pruning, where layers have different pruning ratios [Morcos et al., 2019, Han et al., 2015, Mocanu et al., 2018, Evci et al., 2020, Lee et al., 2021, Xu et al., 2023], compared to uniform pruning [Zhu and Gupta, 2017, Gale et al., 2019]. In Han et al. [2015], Mocanu et al. [2018], Evci et al. [2020], layer-adaptive pruning was achieved by setting a specific magnitude threshold or target pruning ratio for each layer. In Morcos et al. [2019], Lee et al. [2021], layer-adaptive pruning was performed using a global pruning criterion and simultaneously comparing scores from different layers under the target pruning ratio. Specifically, Lee et al. [2021] proposed a global pruning criterion designed from the Frobenius norm-based upper bound of the worst-case $\ell_2$ distortion caused by pruning one layer while fixing the other. Xu et al. [2023] advanced this approach into joint optimization for the sum of filtered layer-wise worst-case $\ell_2$ distortion over pruning ratios. Inspired by Lee et al. [2021], we provide the first global pruning criterion for SSMs, where a *non-magnitude-based* criterion is essential due to the different transfer functions of SSMs compared to other neural networks. Lastly, we provide a missing design motivation for the squaring operation in score evaluation in Lee et al. [2021] by offering a clear rationale based on signal energy.

# 5 Experiments

Table 1 presents the average performance of pruning methods for 10 tasks and 2 models. Further experimental details are explained below.

**Models and tasks.** Experiments were conducted with a single A6000 48GB or RTX 3090 24GB GPU. We verify our method on S4D (S4D-LegS) [Gu et al., 2022a] and S5 [Smith et al., 2023] models, which are multi-SISO and MIMO SSMs, respectively. Although our main motivation was to reduce the state dimension of MIMO models, we also investigated the compressibility of multi-SISO models and the applicability of LAST to them.

Table 1: Average pruning ratio and accuracy loss for all tasks. Values in parentheses are evaluated by excluding non-compressible cases.

| | Model | Avg. prun. ratio (Compressible only) | Avg. accuracy loss ↓ (Compressible only) |
|---|---|---|---|
| | Uniform | 25.00 (33.33) | 0.39 (0.52) |
| S4D | Global | 25.00 (33.33) | 1.16 (1.55) |
| | LAST | 25.00 (33.33) | **0.32 (0.42)** |
| | Uniform | 33.00 (36.67) | 4.32 (4.80) |
| S5 | Global | 33.00 (36.67) | 7.51 (8.35) |
| | LAST | 33.00 (36.67) | **0.52 (0.58)** |

The models were reproduced with three seeds according to the reported configurations [Gu et al., 2022a, Smith et al., 2023] for the six tasks in LRA benchmark [Tay et al., 2021], the raw speech classification task using Speech Commands dataset [Warden, 2018], and pixel-level image classification tasks using MNIST and CIFAR10 datasets. We evaluated the performance of the full (unpruned) and one-shot pruned models while freezing other parameters not involved with SSM layers. See Appendix C for more experimental details.

**Baselines.** The unique transfer functions of SSMs require the state pruning granularity and $\mathcal{H}_\infty$ norm-based pruning criteria, not simple magnitude-based pruning criteria, as validated in Appendix D.

Table 2: Accuracy of pruned models on LRA tasks. LAST is evaluated at the maximum tested pruning ratio with less than 1% accuracy loss, and other methods were evaluated for the same pruning ratios.

| | Model | ListOps (2,048) | | Text (4,096) | | Retrieval (4,000) | | Image (1,024) | | Pathfinder (1,024) | | Path-X (16,384) | | Avg. Acc. |
|---|---|---|---|---|---|---|---|---|---|---|---|---|---|---|
| | | Prun. | Acc. | Prun. | Acc. | Prun. | Acc. | Prun. | Acc. | Prun. | Acc. | Prun. | Acc. | |
| S4D | Full model | 0% | 56.42 | 0% | 86.40 | 0% | 90.46 | 0% | 77.02 | 0% | 87.94 | 0% | 88.07 | 81.05 |
| | Uniform $\mathcal{H}_\infty$ | 10% | 55.82 | 80% | 86.02 | 60% | **89.87** | 0% | 77.02 | 10% | 87.59 | 0% | 88.07 | 80.73 |
| | Global $\mathcal{H}_\infty$ | 10% | 49.95 | 80% | **86.20** | 60% | 89.84 | 0% | 77.02 | 10% | 87.20 | 0% | 88.07 | 79.71 |
| | LAST | 10% | **56.27** | 80% | 85.95 | 60% | 89.46 | 0% | 77.02 | 10% | **87.83** | 0% | 88.07 | **80.77** |
| S5 | Full model | 0% | 61.48 | 0% | 88.88 | 0% | 91.20 | 0% | 87.30 | 0% | 95.15 | 0% | 98.41 | 87.09 |
| | Uniform $\mathcal{H}_\infty$ | 0% | 61.48 | 60% | 82.49 | 50% | 90.29 | 30% | 86.45 | 30% | 71.38 | 30% | 90.90 | 75.50 |
| | Global $\mathcal{H}_\infty$ | 0% | 61.48 | 60% | **88.56** | 50% | **90.93** | 30% | **87.04** | 30% | 57.20 | 30% | 69.21 | 75.74 |
| | LAST | 0% | 61.48 | 60% | 88.52 | 50% | 90.42 | 30% | 86.34 | 30% | **94.45** | 30% | **97.95** | **86.53** |

Here, we compare LAST with two pruning methods: Uniform $\mathcal{H}_\infty$ and Global $\mathcal{H}_\infty$. Uniform $\mathcal{H}_\infty$ utilizes the local pruning criterion, $\mathcal{H}_\infty$ score, and applies the same pruning ratio to each layer. Global $\mathcal{H}_\infty$ employs $\mathcal{H}_\infty$ score as a global criterion, serving as the ablation of the energy normalization used in LAST. Moreover, we present random state pruning results to demonstrate the effectiveness of developed local and global pruning criteria in identifying insignificant states. After one-shot pruning, we evaluate whether these methods appropriately identify significant and insignificant states by measuring accuracy without retraining.

**Pruning ratios.** For models pruned by Global $\mathcal{H}_\infty$ or LAST, which apply layer-adaptive pruning ratios, the reported pruning ratios indicate the *average* pruning ratios across all layers. We compare Uniform $\mathcal{H}_\infty$ and layer-adaptive pruning methods, Global $\mathcal{H}_\infty$ and LAST, by setting the same desired compression rate. The tested pruning ratios were 10%, 20%, $\cdots$, 90%, and 100%, where a pruning ratio of 100% indicates the extreme case leaving only one pair of complex-conjugate subsystems in each layer.

### 5.1 Long range arena

The LRA benchmark [Tay et al., 2021] has been used to evaluate the ability to capture long-range context from sequences with lengths ranging from 1,024 to 16,384. Table 2 shows the accuracy of models when each model is compressed to the maximum pruning ratio at which LAST achieved an accuracy loss below 1% for LRA tasks.

Without any retraining, LAST outperformed other methods, achieving the average accuracy loss of 0.56% (0.67%) for the average compression rate of 33.3% (40.0%), with (without) the non-compressible cases. Since the complexity and state dimension differ across tasks, achievable compression rate varied: for the most compressible case Text, 80% compression on S4D resulted in less than 1% loss in accuracy, while for the least compressible case ListOps, where the state dimension was initially set to 16 for S5 layers, even 10% compression led to large performance degradation.

Figure 2 shows the accuracies of S5 models at different pruning ratios, including randomly pruned models. For Pathfinder and Path-X tasks, LAST consistently outperformed Uniform $\mathcal{H}_\infty$ and Global $\mathcal{H}_\infty$, and the accuracy of Global $\mathcal{H}_\infty$ significantly dropped at high pruning ratios in these cases.

Moreover, we observed that the performance of Uniform $\mathcal{H}_\infty$ was comparable to LAST at low pruning ratios, whereas at high pruning ratios, its performance became inferior to LAST. We hypothesize that this was because the number of significant states in a layer was considerably lower than the number of original states. That is, if Uniform $\mathcal{H}_\infty$ pruning begins pruning beyond the lowest proportion of insignificant states in any layer, it can subsequently cause great accuracy degradation. See Appendix E.2 for full results in LRA tasks.

### 5.2 Raw speech classification

The inductive bias and CT parameterization of SSMs enable 1) encoding raw speech without requiring feature engineering using methods such as short-time Fourier transform and 2) adapting to changes in sampling rate [Gu et al., 2022b, Goel et al., 2022, Gu et al., 2022a, Smith et al., 2023].

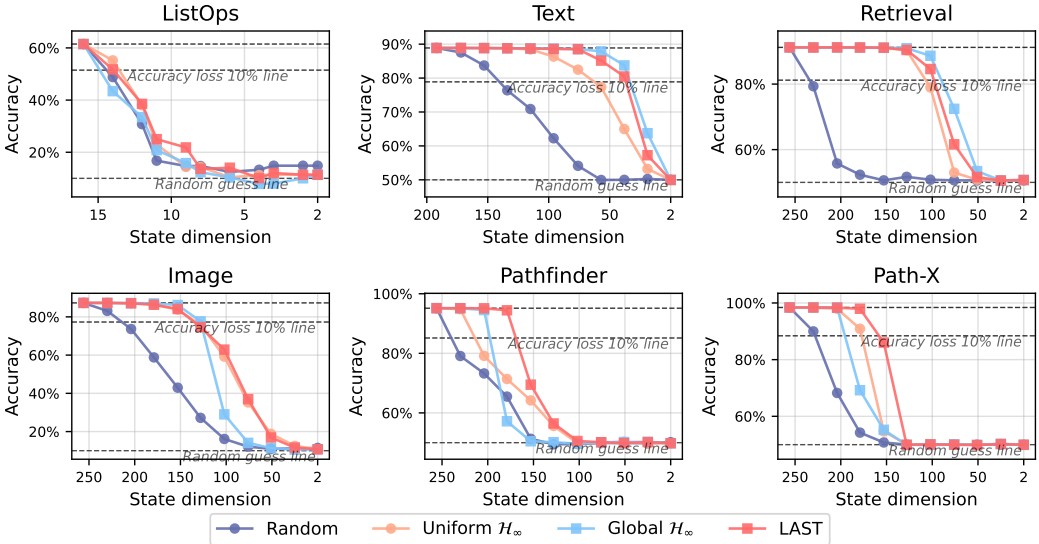

Figure 2: Efficiency-accuracy trade-off curves of pruned S5 models for tasks in LRA benchmark. LAST maintained accuracy better than other methods, Uniform $\mathcal{H}_\infty$ and Global $\mathcal{H}_\infty$ (LAST without energy normalization), demonstrating its superior ability to identify insignificant states.

Table 10 presents that these properties remained consistent after pruning, as pruned models maintained their performance on raw speech and flexibly processed to sequences at different sampling rates by adjusting the learned timescales according to the sampling shifts, similarly to Gu et al. [2022b]. See Figure 8 for more results for Speech Command task.

### 5.3  Pixel-level image classification

We applied pruning to tasks that classify sequenced images, including sequential MNIST (sMNIST), permuted sequential MNIST (psMNIST), and sequential CIFAR (sCIFAR), where sCIFAR is the colored version of Image task in LRA. LAST consistently exhibited the smallest accuracy loss on average. See Appendix E.1 for results on the pixel-level classification tasks.

### 5.4  Analysis

#### 5.4.1  Ablation study on energy normalization

We conduct an ablation study on the energy normalization of LAST by Global $\mathcal{H}_\infty$, which is LAST without using energy normalization. LAST normalizes the differences in layer-wise signal amplification, enabling the cross-layer comparison of states on a common scale. Figure 3 shows the effect of the normalization in S5 models for Path-X task. In Layers 5 and 6, the overall $\mathcal{H}_\infty$ scores were relatively lower than other layers except Layer 1. Global $\mathcal{H}_\infty$ directly used $\mathcal{H}_\infty$ scores, resulting in excessive pruning in Layers 5 and 6 from the pruning ratio of 40%. However, LAST adjusted the scores by accounting for the low total energy transmission of the layers, making their states less prioritized in pruning. This led to different accuracy loss of methods as shown in Figure 2.

Moreover, energy normalization, which excludes pruned subsystems and normalizes accordingly, expands the range of high scores compared to normalizing without exclusions. In the case of Layer 1, this effect results in greater differences between LAST scores, making the scores distinguishable and pruning decisions easier. As a result, Layer 1 was identified to have more insignificant scores compared to other layers, leading to the removal of a large number of states. In conclusion, energy normalization was critical in the pruning process, ensuring a robust cross-layer comparison and preserving the model performance.

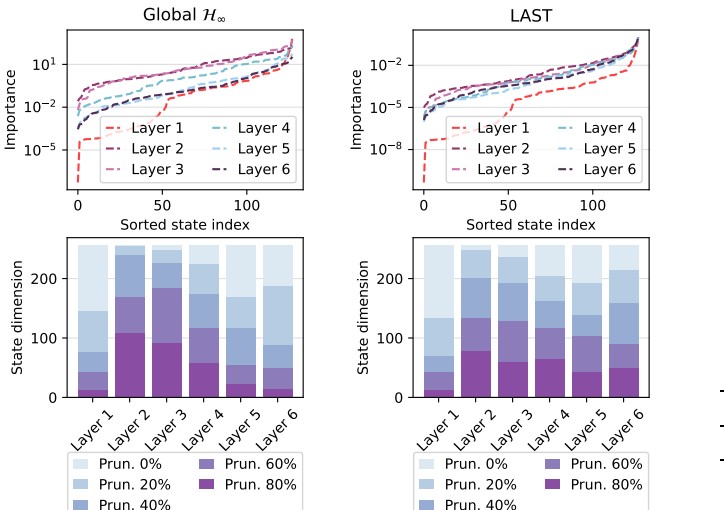

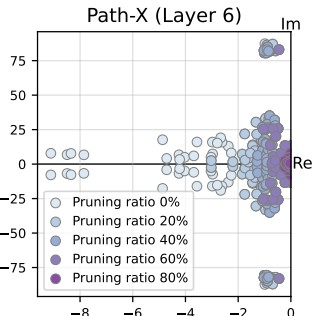

Figure 3: **(Top)** Evaluated state importance score and **(Bottom)** remaining state dimensions in an S5 model for `Path-X` task. The state indices are sorted by $\mathcal{H}_\infty$ scores, evaluated once for each conjugate pair.

Figure 4: Remaining poles in $\mathbf{\Lambda}^{(6)}$ of an S5 model for `Path-X` task.

### 5.4.2 Compressibility of models

We considered S4D, a multi-SISO model, as the equivalent block diagonal MIMO model and applied the same pruning methods. While per-state structured pruning can completely remove state parameters, we implemented masking following the common practice in neural network pruning experiments. This approach allowed us to prune without compromising the parallelism of the multi-SISO model.

We observed that although the effective state dimension is larger in multi-SISO models, the average pruning ratio that does not result in severe accuracy loss was smaller in multi-SISO models (25%) compared to MIMO (33%) models. This is likely because, in multi-SISO, specific states are assigned to specific channels, meaning that each state is given a certain role. Consequently, pruning a single state can result in a greater loss. Additionally, this characteristic resulted in each subsystem exhibiting a significantly low $\mathcal{H}_\infty$ norm.

## 6 Discussion

**Toward efficient training with guaranteed stability.** Relying on guaranteed stability, previous diagonal SSMs have used as many state dimensions as possible due to the challenge of optimizing state dimensions for the task. Although excessive states are initially used, efficient and stable learning can be achieved if insignificant states are pruned under a well-planned pruning schedule.

Given this objective, we first proposed an SSM pruning method that adaptively reduces the order of multiple systems within a deep layered structure with nonlinearity. We derived a local pruning criterion considering the nonlinearity and layer-level output energy loss, applying the criterion in the Uniform $\mathcal{H}_\infty$ and Global $\mathcal{H}_\infty$ methods, which can also be viewed as independently applying traditional MOR to systems in each layer. However, we empirically verified that our method can be more robust than locally applying MOR, particularly when there are significant differences in the $\mathcal{H}_\infty$ norm scale or the proportion of important states across layers. As demonstrated in our application of the proposed method in multi-SISO models, this approach can be applied alongside parallelism.

**Which states are pruned? Lessons for future work.** We investigated the pruned states, which have been judged insignificant for the task, presenting some insightful observations for future work. It is known that $\text{Re}\{\lambda_i\}$ controls the decay rate and $\text{Im}\{\lambda_i\}$ controls the oscillating frequencies of dynamics [Gu et al., 2022a, Chen, 1984]. As the $\mathcal{H}_\infty$ norm is computed with these values, large

$|\text{Re}\{\lambda_i\}|$ (fast decaying mode) and large $|\text{Im}\{\lambda_i\}|$ (high-frequency dynamics) were more prone to be pruned, as shown in Figure 4.

Based on the insignificant pole characteristics, future work might explore new training strategies for SSMs, e.g., making poles constrained to *avoid* having the insignificant characteristics. Moreover, this provides a conjecture for the empirical effectiveness of the block-diagonal initialization in S5 [Smith et al., 2023], suggesting that the initialization performed well because it resulted in fewer large $|\text{Im}\{\lambda_i\}|$. Even using the block-diagonal initialization, we found that previous models tend to have very large $|\text{Im}\{\lambda_i\}|$, e.g., over 1,000, which also could be addressed in future work.

**Limitations.** This paper has the following limitations. Although we explored the pruning criterion for SSMs, questions about when and how often to prune SSMs remained unresolved. Additionally, our proposed method was verified on a specific set of tasks, where both mult-SISO and MIMO models have been evaluated in previous work, and the adaptation to other tasks remains to be investigated. However, we believe that our work opens opportunities to utilize the full capacity of MIMO SSMs by making them as compact as possible, not sacrificing their capacity, search space, or stability.

## Acknowledgments and Disclosure of Funding

This research was supported by the Basic Science Research Program through the National Research Foundation of Korea (NRF) funded by the Ministry of Science, ICT, and Future Planning (2020R1A2C2005709). This work was supported by Samsung Electronics Co., Ltd. (IO201211-08100-01).

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

# Supplementary Material

## Contents

# A  Stability of state space models

## A.1  Indirect pole training

The rational transfer function of an $n$th-order system can be defined by:

$$H(z) = h_0 + \frac{b_1 z^{-1} + \cdots + b_n z^{-n}}{1 + a_1 z^{-1} + \cdots + a_n z^{-n}},$$

following the notation in Parnichkun et al. [2024]. For a second-order system, the characteristic function that determines stability is

$$a(z) = z^2 + a_1 z + a_2.$$

For a DT system to be stable, the poles of the transfer function should be within the unit circle. This can be checked through the Schur-Cohn test [Kailath, 1980], which provides the stable region for the second-order system with the characteristic function $a(z)$ as:

$$a_2^2 < 1 \quad \text{and} \quad (1 + a_2)^2 - a_1^2 > 0,$$

as shown in Figure 5.

### A.1.1  Sufficient stability condition

In models where the poles are trained indirectly, stability can be ensured by applying sufficient constraints for stable poles during training. Montel's constraint [Horn and Johnson, 2012] serves as a sufficient stability condition by restricting the coefficients as follows:

$$\sum_{i=1}^{n} |a_i| \leq 1.$$

For the second-order case, Montel's constraint is

$$|a_1| + |a_2| \leq 1.$$

This defines a sufficient stable region shown in Figure 5. However, as highlighted in Parnichkun et al. [2024], this search space restriction can confine the model performance.

### A.1.2  Stability guaranteed only at initialization

As an alternative, zero initialization [Parnichkun et al., 2024] initializes the system at the center point of the stable region. Thus, the initial coefficients of zero initialization for a second-order system are

$$a_1 = 0 \quad \text{and} \quad a_2 = 0,$$

as marked in Figure 5. However, this does not guarantee stability in subsequent training, which potentially causes states to diverge and makes training infeasible.

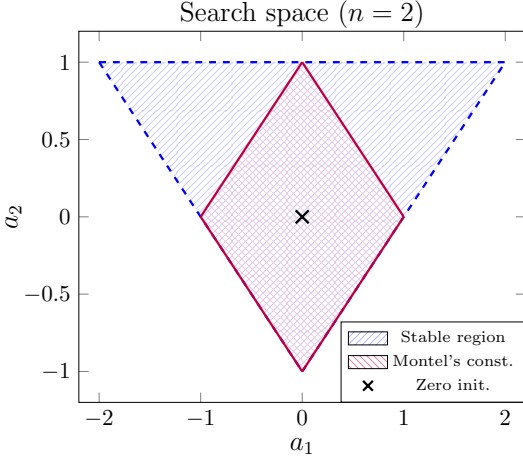

Figure 5: Search space in the two-dimensional coefficient space for stability.

## A.2 Direct pole training

For models like diagonal SSMs that train poles as parameters, it is possible to directly control them to satisfy stability conditions. For example, in the case of CT SSMs, ensuring that the system is Hurwitz, i.e., $\text{Re}(\lambda_i) < 0$ for $i \in \mathcal{S}$, guarantees stability. If a CT SSM is stable, the ZOH-discretized SSM is also stable, i.e., $|\bar{\lambda}_i| < 1$ for $i \in \mathcal{S}$, since

$$
\begin{aligned}
|\bar{\lambda}_i| &= |e^{\lambda_i \Delta_i}| \\
&= |e^{\text{Re}(\lambda_i \Delta_i)} e^{j\text{Im}(\lambda_i \Delta_i)}| \\
&= |e^{\text{Re}(\lambda_i \Delta_i)}||e^{j\text{Im}(\lambda_i \Delta_i)}| \\
&= e^{\text{Re}(\lambda_i \Delta_i)} \\
&< 1,
\end{aligned}
\tag{11}
$$

where the inequality holds since $\text{Re}(\lambda_i) < 0$ and $\Delta_i > 0$ for a stable CT SSM.

## B  Proofs

### B.1  System norm property

The transfer function matrix $\mathbf{G}$ of a system $\Sigma : \mathbf{u} \mapsto \mathbf{y}$ is defined by $\mathbf{Y} = \mathbf{G}\mathbf{U}$, where $\mathbf{U}$ and $\mathbf{Y}$ are the Z-transforms of $\mathbf{u}$ and $\mathbf{y}$. The energy of the output signal $\mathbf{y}$ is bounded with the $\mathcal{H}_\infty$ norm of the system as follows:

$$
\|\mathbf{y}\|_2^2 = \|\mathbf{Y}\|_2^2 \tag{12}
$$

$$
\begin{aligned}
&= \|\mathbf{G}\mathbf{U}\|_2^2 \\
&= \frac{1}{2\pi} \int_0^{2\pi} \|\mathbf{G}(e^{j\theta})\mathbf{U}(e^{j\theta})\|^2 d\theta \\
&\leq \frac{1}{2\pi} \int_0^{2\pi} \|\mathbf{G}(e^{j\theta})\|^2 \|\mathbf{U}(e^{j\theta})\|^2 d\theta \\
&\leq \sup_{\theta \in [0,2\pi]} \bar{\sigma}^2(\mathbf{G}(e^{j\theta})) \left( \frac{1}{2\pi} \int_0^{2\pi} \|\mathbf{U}(e^{j\theta})\|^2 d\theta \right) \\
&= \|\mathbf{G}\|_\infty^2 \|\mathbf{u}\|_2^2,
\end{aligned}
\tag{13}
$$

where we use Parseval's theorem, i.e., $\|\mathbf{v}\|_2^2 = \|\mathcal{Z}(\mathbf{v})\|_2^2$ for a signal $\mathbf{v}$ and the Z-transform $\mathcal{Z}$, in (12) and (13).

### B.2  Bounded layer-level output energy loss

We first show that the TFM of a diagonal system is the sum of the TFMs of subsystems. By applying the Z-transform to (1) (For simplicity, the feed-forward matrix $\mathbf{D}$ is excluded.), we have

$$
z\mathbf{X}(z) = \overline{\mathbf{\Lambda}}\mathbf{X}(z) + \overline{\mathbf{B}}\mathbf{U}(z), \qquad \mathbf{Y}(z) = \mathbf{C}\mathbf{X}(z),
$$

where $\mathbf{X}$, $\mathbf{U}$, and $\mathbf{Y}$ are the Z-transforms of $\mathbf{x}$, $\mathbf{u}$, and $\mathbf{y}$, respectively. We can combine the equations by $\mathbf{Y}(z) = \mathbf{G}(z)\mathbf{U}(z)$, where

$$
\mathbf{G}(z) = \mathbf{C}(z\mathbf{I}_n - \overline{\mathbf{\Lambda}})^{-1}\overline{\mathbf{B}} \tag{14}
$$

$$
= \sum_{i=1}^{n} \frac{\mathbf{C}_i \overline{\mathbf{B}}_i}{z - \bar{\lambda}_i}, \tag{15}
$$

and the decomposition in (15) holds since the considered system is a diagonal system.

Similarly, by applying the Z-transform to subsystem $\Sigma_i$, we can derive its TFM $\mathbf{G}_i$ as follows:

$$
\mathbf{G}_i(z) = \frac{\mathbf{C}_i \overline{\mathbf{B}}_i}{z - \bar{\lambda}_i}. \tag{16}
$$

Substituting (15) with (16) shows that the TFM of the diagonal system is the sum of TFMs of all subsystems as follows:

$$\mathbf{G}(z) = \sum_{i=1}^{n} \mathbf{G}_i(z). \tag{17}$$

Now, we consider the layer-layer output energy loss caused by pruning states in $\mathcal{P}$, i.e., reducing $\Sigma$ into $\Sigma_{-\mathcal{P}}$. In previous SSMs, GELU [Hendrycks and Gimpel, 2016] has been widely used as the activation function. Using the 1-Lipschitzness of GELU, TFM decomposition (17), and $\mathcal{H}_\infty$ norm property (2), the layer-level energy loss is upper bounded as follows:

$$
\begin{aligned}
\|f_\sigma(\mathbf{u}; \Sigma) - f_\sigma(\mathbf{u}; \Sigma_{-\mathcal{P}})\|_2^2 &= \|\sigma(\Sigma(\mathbf{u})) - \sigma(\Sigma_{-\mathcal{P}}(\mathbf{u}))\|_2^2 \\
&\leq \|\Sigma(\mathbf{u}) - \Sigma_{-\mathcal{P}}(\mathbf{u})\|_2^2 && \because \text{1-Lipschitzness} \\
&= \|\mathbf{G}\mathbf{U} - \mathbf{G}_{-\mathcal{P}}\mathbf{U}\|_2^2 && \because \text{Parseval's theorem,} \\
& && \quad \text{Linearity of Z-transform} \\
&= \|\sum_{i \in \mathcal{P}} \mathbf{G}_i \mathbf{U}\|_2^2 && \because (17) \\
&\leq \sum_{i \in \mathcal{P}} \|\mathbf{G}_i \mathbf{U}\|_2^2 && \because \text{Triangle inequality} \\
&= \sum_{i \in \mathcal{P}} \|\Sigma_i(\mathbf{u})\|_2^2 && \because \text{Parseval's theorem} \\
&\leq \sum_{i \in \mathcal{P}} \|\mathbf{G}_i\|_\infty^2 \|\mathbf{u}\|_2^2. && \because (2)
\end{aligned}
$$

This inequality builds upon the approach in Neyshabur et al. [2015], Lee et al. [2021], adapting it for diagonal SSMs and deriving bounds using signal and system norms. In the above derivation, we can analyze on a subsystem basis by utilizing the diagonal structure. Parseval's theorem [Green and Limebeer, 2012] allows us to switch between the time and frequency domains with energy equivalence. Even in the presence of nonlinear functions and signal distortion analysis, we achieved a result similar to modal truncation [Green and Limebeer, 2012], where the $\mathcal{H}_\infty$ norm distortion of an LTI system is bounded by the sum of the $\mathcal{H}_\infty$ norms of the truncated LTI systems.

### B.3 Bounded model-level output energy loss

We show that the model-level output energy loss in Equation (8) is bounded with $\mathcal{H}_\infty$ norms of subsystems:

$$
J_l\big(i; \widetilde{\Sigma}_t^{(1:L)}\big) \leq \frac{\|\mathbf{G}_i^{(l)}\|_\infty^2}{\|\mathbf{G}_{\mathcal{S}_t^{(l)}}^{(l)}\|_\infty^2} \prod_{k=1}^{L} \|\mathbf{G}_{\mathcal{S}_t^{(k)}}^{(k)}\|_\infty^2 \|\mathbf{u}^{(1)}\|_2^2,
$$

where $J_l\big(i; \widetilde{\Sigma}_t^{(1:L)}\big) := \left\| f_\sigma(\mathbf{u}^{(1)}; \widetilde{\Sigma}_t^{(1:L)}) - f_\sigma(\mathbf{u}^{(1)}; \widetilde{\Sigma}_t^{(1:l-1)}, \Sigma_{\mathcal{S}_t^{(l)} \setminus \{i\}}^{(l)}, \widetilde{\Sigma}_t^{(l+1:L)}) \right\|_2^2.$

From $L$th layer to $l+1$th layer, we can keep bounding the output energy of formal layers using the 1-Lipschitzness of $\sigma(\cdot)$ and $\mathcal{H}_\infty$ norm property in Equation (2) as follows:

$$
\begin{aligned}
& \left\| f_\sigma(\mathbf{u}^{(1)}; \widetilde{\Sigma}_t^{(1:L)}) - f_\sigma(\mathbf{u}^{(1)}; \widetilde{\Sigma}_t^{(1:l-1)}, \Sigma_{\mathcal{S}_t^{(l)} \setminus \{i\}}^{(l)}, \widetilde{\Sigma}_t^{(l+1:L)}) \right\|_2^2 \\
&= \left\| \sigma(\Sigma_{\mathcal{S}_t^{(L)}}^{(L)}(\mathbf{u}^{(1)}; \widetilde{\Sigma}_t^{(1:L-1)})) - \sigma(\Sigma_{\mathcal{S}_t^{(L)}}^{(L)}(\mathbf{u}^{(1)}; \widetilde{\Sigma}_t^{(1:l-1)}, \Sigma_{\mathcal{S}_t^{(l)} \setminus \{i\}}^{(l)}, \widetilde{\Sigma}_t^{(l+1:L-1)})) \right\|_2^2 \\
&\leq \left\| \Sigma_{\mathcal{S}_t^{(L)}}^{(L)}(\mathbf{u}^{(1)}; \widetilde{\Sigma}_t^{(1:L-1)}) - \Sigma_{\mathcal{S}_t^{(L)}}^{(L)}(\mathbf{u}^{(1)}; \widetilde{\Sigma}_t^{(1:l-1)}, \Sigma_{\mathcal{S}_t^{(l)} \setminus \{i\}}^{(l)}, \widetilde{\Sigma}_t^{(l+1:L-1)}) \right\|_2^2 \\
&\leq \left\| \mathbf{G}_{\mathcal{S}_t^{(L)}}^{(L)} \right\|_\infty^2 \left\| f_\sigma(\mathbf{u}^{(1)}; \widetilde{\Sigma}_t^{(1:L-1)}) - f_\sigma(\mathbf{u}^{(1)}; \widetilde{\Sigma}_t^{(1:l-1)}, \Sigma_{\mathcal{S}_t^{(l)} \setminus \{i\}}^{(l)}, \widetilde{\Sigma}_t^{(l+1:L-1)}) \right\|_2^2
\end{aligned}
$$

$$\vdots$$

$$\leq \prod_{k=l+1}^{L} \left\| \mathbf{G}_{\mathcal{S}_t^{(k)}}^{(k)} \right\|_{\infty}^2 \left\| \widetilde{\Sigma}_t^{(l)} \left( f_\sigma(\mathbf{u}^{(1)}; \widetilde{\Sigma}_t^{(1:l-1)}) \right) - \Sigma_{\mathcal{S}_t^{(l)} \setminus \{i\}}^{(l)} \left( f_\sigma(\mathbf{u}^{(1)}; \widetilde{\Sigma}_t^{(1:l-1)}) \right) \right\|_2^2.$$

Since $\widetilde{\Sigma}_t^{(l)} := \Sigma_{\mathcal{S}_t^{(l)}}^{(l)}$, we can apply the bounded layer-level energy loss property in Equation (4):

$$\prod_{k=l+1}^{L} \left\| \mathbf{G}_{\mathcal{S}_t^{(k)}}^{(k)} \right\|_{\infty}^2 \left\| \widetilde{\Sigma}_t^{(l)} \left( f_\sigma(\mathbf{u}^{(1)}; \widetilde{\Sigma}_t^{(1:l-1)}) \right) - \Sigma_{\mathcal{S}_t^{(l)} \setminus \{i\}}^{(l)} \left( f_\sigma(\mathbf{u}^{(1)}; \widetilde{\Sigma}_t^{(1:l-1)}) \right) \right\|_2^2$$

$$= \prod_{k=l+1}^{L} \left\| \mathbf{G}_{\mathcal{S}_t^{(k)}}^{(k)} \right\|_{\infty}^2 \left\| \Sigma_{\mathcal{S}_t^{(l)}}^{(l)} \left( f_\sigma(\mathbf{u}^{(1)}; \widetilde{\Sigma}_t^{(1:l-1)}) \right) - \Sigma_{\mathcal{S}_t^{(l)} \setminus \{i\}}^{(l)} \left( f_\sigma(\mathbf{u}^{(1)}; \widetilde{\Sigma}_t^{(1:l-1)}) \right) \right\|_2^2$$

$$\leq \left\| \mathbf{G}_i^{(l)} \right\|_{\infty}^2 \prod_{k=l+1}^{L} \left\| \mathbf{G}_{\mathcal{S}_t^{(k)}}^{(k)} \right\|_{\infty}^2 \left\| f_\sigma(\mathbf{u}^{(1)}; \widetilde{\Sigma}_t^{(1:l-1)}) \right\|_2^2.$$

Then we can add an auxiliary term $\sigma(0)$ and use the 1-Lipschitzness property since $\sigma(0) = 0$ holds for GELU activation $\sigma$:

$$\left\| \mathbf{G}_i^{(l)} \right\|_{\infty}^2 \prod_{k=l+1}^{L} \left\| \mathbf{G}_{\mathcal{S}_t^{(k)}}^{(k)} \right\|_{\infty}^2 \left\| f_\sigma(\mathbf{u}^{(1)}; \widetilde{\Sigma}_t^{(1:l-1)}) \right\|_2^2$$

$$= \left\| \mathbf{G}_i^{(l)} \right\|_{\infty}^2 \prod_{k=l+1}^{L} \left\| \mathbf{G}_{\mathcal{S}_t^{(k)}}^{(k)} \right\|_{\infty}^2 \left\| \sigma(\widetilde{\Sigma}_t^{(l-1)}(f_\sigma(\mathbf{u}^{(1)}; \widetilde{\Sigma}_t^{(1:l-2)}))) - \sigma(0) \right\|_2^2$$

$$\leq \left\| \mathbf{G}_i^{(l)} \right\|_{\infty}^2 \prod_{k=l+1}^{L} \left\| \mathbf{G}_{\mathcal{S}_t^{(k)}}^{(k)} \right\|_{\infty}^2 \left\| \widetilde{\Sigma}_t^{(l-1)}(f_\sigma(\mathbf{u}^{(1)}; \widetilde{\Sigma}_t^{(1:l-2)})) \right\|_2^2.$$

Again, the output of $l-1$th layer can be bounded using the $\mathcal{H}_\infty$ norm property, and keeping the procedures to the first layer proves the statement as follows:

$$\left\| \mathbf{G}_i^{(l)} \right\|_{\infty}^2 \prod_{k=l+1}^{L} \left\| \mathbf{G}_{\mathcal{S}_t^{(k)}}^{(k)} \right\|_{\infty}^2 \left\| \widetilde{\Sigma}_t^{(l-1)}(f_\sigma(\mathbf{u}^{(1)}; \widetilde{\Sigma}_t^{(1:l-2)})) \right\|_2^2$$

$$\leq \left\| \mathbf{G}_i^{(l)} \right\|_{\infty}^2 \prod_{k=l+1}^{L} \left\| \mathbf{G}_{\mathcal{S}_t^{(k)}}^{(k)} \right\|_{\infty}^2 \left\| \mathbf{G}_{\mathcal{S}_t^{(l-1)}}^{(l-1)} \right\|_{\infty}^2 \left\| f_\sigma(\mathbf{u}^{(1)}; \widetilde{\Sigma}_t^{(1:l-2)}) \right\|_2^2$$

$$\vdots$$

$$\leq \left\| \mathbf{G}_i^{(l)} \right\|_{\infty}^2 \prod_{k=l+1}^{L} \left\| \mathbf{G}_{\mathcal{S}_t^{(k)}}^{(k)} \right\|_{\infty}^2 \prod_{k=l}^{l-1} \left\| \mathbf{G}_{\mathcal{S}_t^{(k)}}^{(k)} \right\|_{\infty}^2 \left\| \mathbf{u}^{(1)} \right\|_2^2$$

$$= \frac{\left\| \mathbf{G}_i^{(l)} \right\|_{\infty}^2}{\left\| \mathbf{G}_{\mathcal{S}_t^{(l)}}^{(l)} \right\|_{\infty}^2} \prod_{k=1}^{L} \left\| \mathbf{G}_{\mathcal{S}_t^{(k)}}^{(k)} \right\|_{\infty}^2 \left\| \mathbf{u}^{(1)} \right\|_2^2.$$

## C  Experimental details

Our experiments were conducted with JAX [Bradbury et al., 2018] on a single A6000 48GB GPU or RTX 3090 24GB GPU. We reproduced S4D models using the implementations from Rush and Karamcheti [2022][2] and S5 models from Smith et al. [2023][3]. We used bidirectional SSMs for all

[2] https://github.com/srush/annotated-s4
[3] https://github.com/lindermanlab/S5

tasks except `sMNIST` and `psMNIST` tasks. Following S5, we implemented bidirectional S4D models to have $\mathbf{C}^b$ matrices for reverse convolution. For inference, we used Vandermonde product and convolution kernel schemes for S4D and parallel scans for S5 models.

## C.1 Tasks

The following lists ten tasks where we tested our proposed method, along with the specified resources and the time taken for model training for each task.

- `sMNIST`: 10-way classification task with flattened MNIST images, each having a sequence length of 784. Original images are for handwritten digits. It took 30 minutes to train an S5 model with an RTX 3090 24GB GPU.

- `psMNIST`: 10-way classification task with flattened and fixed-order permuted MNIST images, each having a sequence length of 784. Original images are for handwritten digits. It took 1 hour to train an S5 model with an RTX 3090 24GB GPU.

- `sCIFAR`: 10-way classification task with flattened CIFAR-10 images [Krizhevsky et al., 2009], each having a sequence length of 1,024 for each R, G, B channel. The dataset includes 45,000 training, 5,000 validation, and 10,000 test sequences. It took 7 hours to train an S4D model or an S5 model with an A6000 48GB GPU.

- `ListOps`: 10-way classification task with longer variations of ListOps data [Nangia and Bowman, 2018], each having a maximum sequence length of 2048 for a single channel. The task is solving nested mathematical operations applied to numbers in the range of 0-9 to derive a final result. One-hot vectors for 17 values, including operators, enclosers of operators, and numbers, are concatenated. The dataset includes 96,000 training, 2,000 validation, and 2,000 test sequences. It took 2 hours to train an S4D model or an S5 model with an RTX 3090 24GB GPU.

- `Text`: 2-way byte-level text classification task with IMDB review data [Maas et al., 2011], each having a maximum sequence length of 4,096 for a single channel. The task is classifying the sentiment of a review. One-hot vectors for 129 characters are concatenated. The dataset includes 25,000 training and 25,000 test sequences. It took 2.5 hours to train an S4D model and 1.5 hours to train an S5 model with an RTX 3090 24GB GPU.

- `Retrieval`: 2-way byte-level document retrieval task with ACL Anthology Network document data [Radev et al., 2009], each having a maximum sequence length of 4,000 for a single channel. The task is classifying if two documents are linked by equivalent citations. One-hot vectors for 97 characters are concatenated for each document. The dataset includes 147,086 training, 18,090 validation, and 17,437 test sequence pairs. It took 15.5 hours to train an S4D model with an A6000 48GB GPU and 6 hours to train an S5 model with an RTX 3090 24GB GPU.

- `Image`: 10-way classification task with flattened CIFAR-10 images [Krizhevsky et al., 2009], each having a sequence length of 1,024 for a single channel. It took 9.5 hours to train an S4D model and 7.5 hours to train an S5 model with an RTX 3090 24GB GPU.

- `Pathfinder`: 2-way classification task with flattened Pathfinder challenge images [Linsley et al., 2018], each having a sequence length of 1,024 for a single channel. Original images are for points with connecting or distracting paths. The dataset includes 160,000 training, 20,000 validation, and 20,000 test sequences. It took 14 hours to train an S4D model and 11 hours to train an S5 model with an RTX 3090 24GB GPU.

- `Path-X`: 2-way classification task with flattened scaled Pathfinder challenge images [Linsley et al., 2018], each having a sequence length of 16,384 for a single channel. Original images are for points with connecting or distracting paths. Original images are for points and connecting or distracting paths. It took 3 days to train an S4D model and 1 day to train an S5 model with an A6000 48GB GPU.

- `Speech Command`: 35-way classification task with 1-second word-speaking audio recording data [Warden, 2018], each having a sequence length of 16,000 for a single channel. For the varying sampling frequency tests, the data was downsampled from 16kHz to 8kHz. The dataset includes 24,482 training, 5,246 validation, and 5,247 test sequences. It took 21 hours to train an S4D model and 8 hours to train an S5 model with an RTX 3090 24GB GPU.

## C.2 Hyperparameters

We followed the hyperparameters in [Gu et al., 2022a, Smith et al., 2023]. For `Path-X` task, it was challenging to train S4D models with the original learning rate of 0.0005, thus we changed it to 0.001.

Table 3: Training configurations of S4D models for all tested tasks. $n_s$: state dimension of each SISO system. LN: layer normalization, BN: batch normalization, Pre: pre-normalization. D: dropout. LR: learning rate. B: batch size. E: epochs. WD: weight decay. [†]: The value is changed from the original release [Gu et al., 2022a] for training feasibility.

| Task | $L$ | $h$ | $n_s$ | Norm | Pre | D | LR | B | E | WD | $(\Delta_{\min}, \Delta_{\max})$ |
|------|-----|-----|-------|------|-----|---|----|---|---|----|-----------------------------------|
| sCIFAR | 6 | 512 | 64 | LN | False | 0.1 | 0.01 | 50 | 200 | 0.05 | (0.001, 0.1) |
| ListOps | 8 | 128 | 64 | BN | False | 0 | 0.01 | 50 | 40 | 0.05 | (0.001, 0.1) |
| Text | 6 | 256 | 64 | BN | True | 0 | 0.01 | 16 | 32 | 0.05 | (0.001, 0.1) |
| Retrieval | 6 | 256 | 64 | BN | True | 0 | 0.01 | 64 | 20 | 0.05 | (0.001, 0.1) |
| Image | 6 | 512 | 64 | LN | False | 0.1 | 0.01 | 50 | 200 | 0.05 | (0.001, 0.1) |
| Pathfinder | 6 | 256 | 64 | BN | True | 0 | 0.004 | 64 | 200 | 0.03 | (0.001, 0.1) |
| Path-X | 6 | 256 | 64 | BN | True | 0 | $0.001^{\dagger}$ | 32 | 50 | 0.05 | (0.001, 0.01) |
| Speech | 6 | 128 | 64 | BN | True | 0 | 0.01 | 16 | 40 | 0.05 | (0.001, 0.1) |

Table 4: Training configurations of S5 models for all tested tasks. All models used batch normalization, pre-normalization, and $\Delta_{max} = 0.1$. $n_m$: state dimension of a MIMO system. $J$: number of blocks for block initialization of $\mathbf{\Lambda}$. D: dropout. LR: learning rate. SSM LR: learning rate for SSM parameters, B: batch size. E: epochs. WD: weight decay.

| Task | $L$ | $h$ | $n_m$ | $J$ | D | LR | SSM LR | B | E | WD | $\Delta_{\min}$ |
|------|-----|-----|-------|-----|---|----|--------|---|---|----|-----------------|
| sMNIST | 4 | 96 | 128 | 1 | 0.1 | 0.008 | 0.002 | 50 | 150 | 0.01 | 0.001 |
| psMNIST | 4 | 128 | 128 | 2 | 0.15 | 0.004 | 0.001 | 50 | 150 | 0.01 | 0.001 |
| sCIFAR | 6 | 512 | 384 | 3 | 0.1 | 0.0045 | 0.001 | 50 | 250 | 0.07 | 0.001 |
| ListOps | 8 | 128 | 16 | 8 | 0 | 0.003 | 0.001 | 50 | 40 | 0.07 | 0.001 |
| Text | 6 | 256 | 192 | 12 | 0.1 | 0.004 | 0.001 | 50 | 35 | 0.07 | 0.001 |
| Retrieval | 6 | 128 | 256 | 16 | 0 | 0.002 | 0.001 | 32 | 20 | 0.05 | 0.001 |
| Image | 6 | 512 | 384 | 3 | 0.1 | 0.005 | 0.001 | 50 | 250 | 0.07 | 0.001 |
| Pathfinder | 6 | 192 | 256 | 8 | 0.05 | 0.005 | 0.0009 | 64 | 200 | 0.07 | 0.001 |
| Path-X | 6 | 128 | 256 | 16 | 0 | 0.002 | 0.0006 | 32 | 75 | 0.05 | 0.001 |
| Speech | 6 | 96 | 128 | 16 | 0.1 | 0.008 | 0.002 | 16 | 40 | 0.04 | 0.001 |

# D Validation of pruning granularity and criterion

## D.1 State pruning granularity

As in channel pruning [He et al., 2017], state pruning is named based on its granularity of pruning, that is, all parameters associated with insignificant states are pruned at once. For instance, the parameters $\lambda_i$ from $\Lambda$, the row vector $\mathbf{B}_i$ from $\mathbf{B}$, and the column vector $\mathbf{C}_i$ from $\mathbf{C}$ are pruned when the state $i$ is identified as an insignificant state.

To explicitly demonstrate the necessity of state pruning in SSMs, we compared the performance of unstructured random pruning and structured random state pruning using S5 models. For unstructured random pruning, we pruned randomly selected elements from the system matrices, obtaining the results in Table 5.

Despite the similar number of parameters being pruned, the model suffered a significant performance degradation, with an average accuracy loss of 59.92%, in the case of unstructured random pruning.

Table 5: Average pruning ratio and accuracy loss for all tasks. Values in parentheses are evaluated by excluding non-compressible cases.

| Method | Average pruning ratio | Average accuracy loss ↓ |
|---|---|---|
| Unstructured random | 33.00 (36.67) | 59.92 (66.58) |
| Structured random | 33.00 (36.67) | **29.53 (32.82)** |

This is because unstructured pruning can alter the learned dynamics in all subsystems. In contrast, state pruning maintains the functionality of unpruned subsystems, leading to less performance degradation. This highlights the importance of considering the structure and mechanism of the model when applying pruning techniques.

### D.2 Comparison with magnitude pruning

Magnitudes and $L_p$ norms of parameters are simple but effective pruning criteria to obtain efficient neural networks [Cheng et al., 2024]. Given the necessity of state pruning granularity in SSMs, we set the pruning granularity to state pruning and then compared the significant state identification abilities of the magnitude and $\mathcal{H}_\infty$ pruning methods. To extend Table 1, we define magnitude state pruning methods as follows:

- **Uniform Magnitude.** Every layer is uniformly pruned to have the same pruning ratio with the importance of each state $i$ as $|\bar{\lambda}_i|\|\overline{\mathbf{B}}_i\|\|\mathbf{C}_i\|$. While any $L_p$ norm can be used, we present the results using the $L_2$ norm as an example.

- **Global Magnitude.** The same state importance criterion as in Uniform Magnitude is used, but the comparison group is extended from intra-layer to inter-layer, ensuring that the pruning ratio is met globally for the entire network.

- **LAMP.** This method employs a criterion of $\frac{|\bar{\lambda}_i|^2\|\overline{\mathbf{B}}_i\|^2\|\mathbf{C}_i\|^2}{\sum_{j\leq i}|\bar{\lambda}_j|^2\|\overline{\mathbf{B}}_j\|^2\|\mathbf{C}_j\|^2}$ adapted from Lee et al. [2021], which originally used $\frac{W_i^2}{\sum_{j\leq i}W_j^2}$ as a criterion for a real-valued weight parameter $W$. The state indices in the denominator are assumed to be ordered based on their evaluation using the basic magnitude criterion similar to LAST.

Extending the S5 model part in Table 1, Table 6 reports that, at the same pruning ratio, LAST and other $\mathcal{H}_\infty$ pruning methods significantly outperform magnitude pruning methods by showing less accuracy loss, which implies that $\mathcal{H}_\infty$ pruning methods can better distinguish significant and insignificant states. This performance gap and suitability can be explained with the unique transfer function of SSMs, which is defined in the frequency domain as in Equation (14) for the whole system and Equation (16) for a subsystem. Specifically, the importance of $\bar{\lambda}_i$ is evaluated by $(1 - |\bar{\lambda}_i|)^{-1}$ in $\mathcal{H}_\infty$ pruning methods, while magnitude pruning methods evaluate $|\bar{\lambda}_i|$.

Table 6: Average pruning ratio and accuracy loss in S5 models for all tasks. Values in parentheses are evaluated by excluding non-compressible cases.

| Method | Average pruning ratio | Average accuracy loss ↓ | State importance |
|---|---|---|---|
| Random | 33.00 (36.67) | 29.53 (32.82) | - |
| Uniform magnitude | 33.00 (36.67) | 22.03 (24.48) | $\lvert\overline{\lambda}_i\rvert\,\lVert\overline{\mathbf{B}}_i\rVert\,\lVert\mathbf{C}_i\rVert$ |
| Global magnitude | 33.00 (36.67) | 17.49 (19.43) | $\lvert\overline{\lambda}_i\rvert\,\lVert\overline{\mathbf{B}}_i\rVert\,\lVert\mathbf{C}_i\rVert$ |
| LAMP | 33.00 (36.67) | 18.07 (20.07) | $\dfrac{\lvert\overline{\lambda}_i\rvert^2\lVert\overline{\mathbf{B}}_i\rVert^2\lVert\mathbf{C}_i\rVert^2}{\sum_{j\leq i}\lvert\overline{\lambda}_j\rvert^2\lVert\overline{\mathbf{B}}_j\rVert^2\lVert\mathbf{C}_j\rVert^2}$ |
| Uniform $\mathcal{H}_\infty$ | 33.00 (36.67) | 4.32 (4.80) | $\dfrac{\lVert\mathbf{C}_i\rVert^2\lVert\overline{\mathbf{B}}_i\rVert^2}{(1-\lvert\overline{\lambda}_i\rvert)^2}$ |
| Global $\mathcal{H}_\infty$ | 33.00 (36.67) | 7.51 (8.35) | $\dfrac{\lVert\mathbf{C}_i\rVert^2\lVert\overline{\mathbf{B}}_i\rVert^2}{(1-\lvert\overline{\lambda}_i\rvert)^2}$ |
| LAST | 33.00 (36.67) | **0.52 (0.58)** | $\dfrac{\frac{\lVert\mathbf{C}_i\rVert^2\lVert\overline{\mathbf{B}}_i\rVert^2}{(1-\lvert\overline{\lambda}_i\rvert)^2}}{\sum_{j\leq i}\frac{\lVert\mathbf{C}_j\rVert^2\lVert\overline{\mathbf{B}}_j\rVert^2}{(1-\lvert\overline{\lambda}_j\rvert)^2}}$ |

# E  Full results

## E.1  Pixel-level image classification

Table 7 highlights the results evaluated at the maximum pruning ratio where the accuracy loss of LAST was less than 1%. Figure 6 shows the accuracy at all tested pruning ratios.

As shown in Table 7 and Figure 6, both S4D and S5 had great compressibility. In Table 7, the state dimension of S4D indicates the average $n_s$ of SISO systems, while in Figure 6, it refers to the average effective state dimension $n$ across layers.

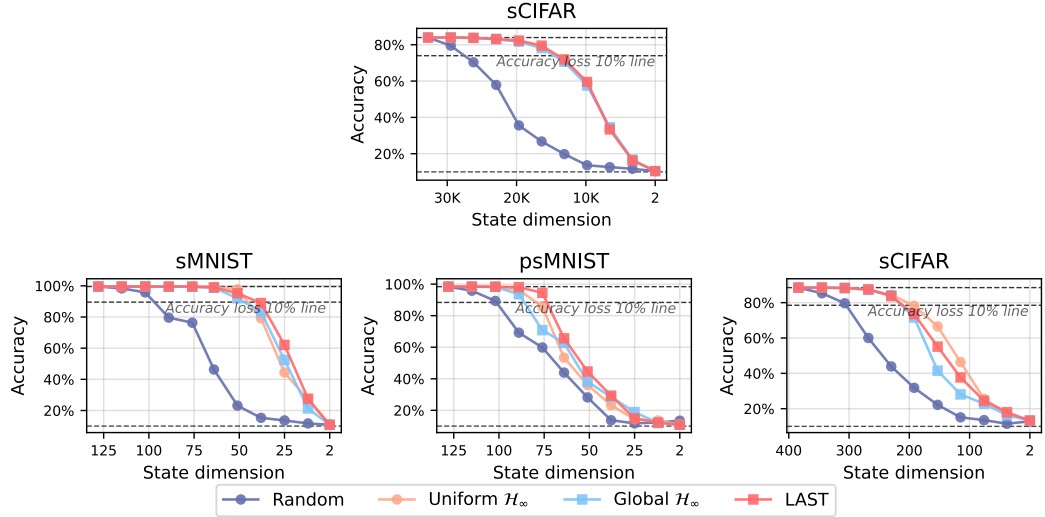

Figure 6: Efficiency-accuracy trade-off curves of pruned **(Upper)** S4D **(Lower)** S5 models for pixel-level image classification tasks. LAST obtained more efficient models that maintain performance compared to Uniform $\mathcal{H}_\infty$, which was observed more stably and consistently than Global $\mathcal{H}_\infty$ (LAST w/o score normalization).

Table 7: Accuracy of pruned models on pixel-level image classification tasks. LAST is evaluated at the maximum tested pruning ratio with less than 1% accuracy loss, and other methods were evaluated for the same pruning ratios.

| Task | Model | Method | Prun. | State dim. | Accuracy |
|------|-------|--------|-------|-----------|----------|
| sMNIST (784) | S4D | - | - | - | - |
| | S5 | Full model | 0% | 128 | $99.55 \pm 0.02$ |
| | | Uniform $\mathcal{H}_\infty$ | 50% | 64 | $\mathbf{99.26 \pm 0.15}$ |
| | | Global $\mathcal{H}_\infty$ | 50% | 64 | $98.75 \pm 0.24$ |
| | | LAST | 50% | 64 | $99.01 \pm 0.57$ |
| psMNIST (784) | S4D | - | - | - | - |
| | S5 | Full model | 0% | 128 | $98.39 \pm 0.06$ |
| | | Uniform $\mathcal{H}_\infty$ | 30% | 90 | $96.32 \pm 0.19$ |
| | | Global $\mathcal{H}_\infty$ | 30% | 90 | $93.59 \pm 3.82$ |
| | | LAST | 30% | 90 | $\mathbf{98.09 \pm 0.30}$ |
| sCIFAR (1,024) | S4D | Full model | 0% | 64 | $83.97 \pm 0.30$ |
| | | Uniform $\mathcal{H}_\infty$ | 30% | 45 | $83.10 \pm 0.35$ |
| | | Global $\mathcal{H}_\infty$ | 30% | 45 | $83.02 \pm 0.46$ |
| | | LAST | 30% | 45 | $\mathbf{83.21 \pm 0.33}$ |
| | S5 | Full model | 0% | 384 | $88.52 \pm 0.29$ |
| | | Uniform $\mathcal{H}_\infty$ | 30% | 269 | $87.37 \pm 0.85$ |
| | | Global $\mathcal{H}_\infty$ | 30% | 269 | $87.22 \pm 0.05$ |
| | | LAST | 30% | 269 | $\mathbf{87.53 \pm 0.41}$ |

## E.2 Long range arena

To evaluate the practical efficiency resulting from LAST, we implemented pruning by removal, in addition to pruning by masking implementation, by transferring selected significant parameters to a smaller-dimensional model. Table 8 presents the average evaluation step speed and peak GPU memory usage of pruned S5 models for an NVIDIA RTX 3090 GPU. Reducing the state dimension improved efficiency in both computational and memory costs, with the degree of efficiency depending on the channel size per task.

Table 8: Efficiency improvement in computational and memory costs in S5 models.

| | ListOps | Text | Retrieval | Image | Pathfinder | Path-X |
|---|---------|------|-----------|-------|------------|--------|
| Pruning ratio | 0% | 60% | 50% | 30% | 30% | 30% |
| Inference speed ↑ | 1.0× | 1.6× | 1.7× | 1.2× | 1.1× | 1.3× |
| GPU memory usage ↓ | 1.0× | 0.9× | 0.6× | 1.0× | 0.8× | 0.8× |

Table 9 highlights the results evaluated at the maximum pruning ratio where the accuracy loss of LAST was less than 1%. Figure 7 shows the accuracy at all tested pruning ratios. In Table 9, the state dimension of S4D indicates the average $n_s$ of SISO systems, while in Figure 7, it refers to the average effective state dimension $n$ across layers.

In ListOps task, where the initial state dimension was small, the S5 models were uncompressible. For Text task, both S4D and S5 models showed the highest compressibility among all tasks, followed by Retrieval task.

In Image task, S4D models were uncompressible since the $\mathcal{H}_\infty$ scores were significantly low and fell below the precision threshold of the floating-point representation, making the comparison in local pruning and sorting required for LAST score calculation impossible.

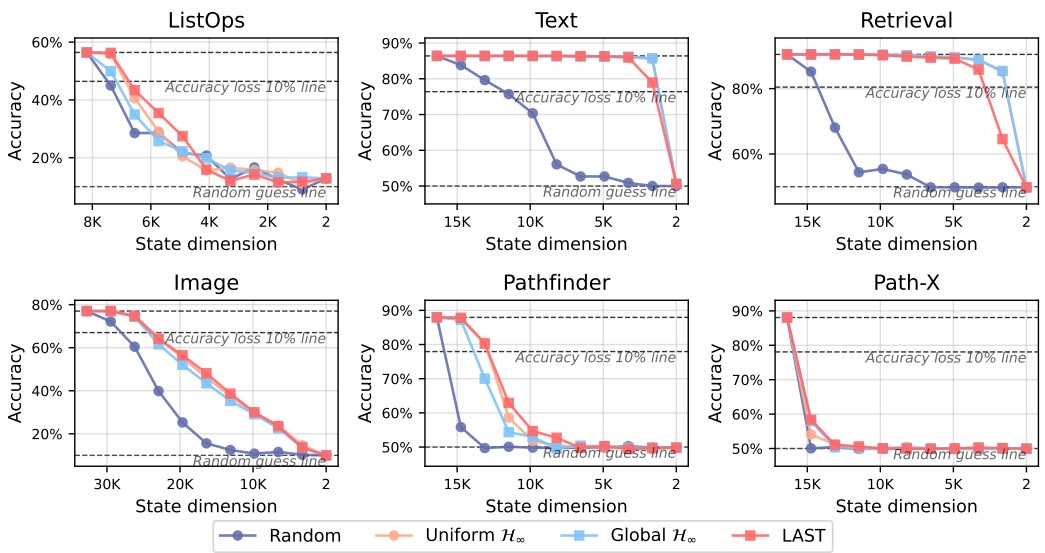

Figure 7: Efficiency-accuracy trade-off curves of pruned S4D models for LRA tasks. LAST obtained more efficient models that maintain performance compared to Uniform $\mathcal{H}_\infty$, which was observed more stably and consistently than Global $\mathcal{H}_\infty$ (LAST w/o score normalization).

Notably, the state dimensions of S5 models were able to reduce by 30% in both the `Pathfinder` and `Path-X` tasks. The ability to maintain performance in `Path-X` highlights the effectiveness of the MIMO structure of S5.

Table 9: Accuracy of pruned models for LRA tasks. LAST is evaluated at the maximum tested pruning ratio with less than 1% accuracy loss, and other methods were evaluated for the same pruning ratios.

| Task | Model | Method | Prun. | State dim. | Accuracy |
|---|---|---|---|---|---|
| ListOps (2,048) | S4D | Full model | 0% | 64 | $56.42 \pm 0.02$ |
| | | Uniform $\mathcal{H}_\infty$ | 10% | 58 | $55.82 \pm 0.81$ |
| | | Global $\mathcal{H}_\infty$ | 10% | 58 | $49.95 \pm 7.32$ |
| | | LAST | 10% | 58 | $\mathbf{56.27 \pm 0.70}$ |
| | S5 | Full model | 0% | 16 | $61.48 \pm 0.24$ |
| | | Uniform $\mathcal{H}_\infty$ | 0% | 16 | $61.48 \pm 0.24$ |
| | | Global $\mathcal{H}_\infty$ | 0% | 16 | $61.48 \pm 0.24$ |
| | | LAST | 0% | 16 | $61.48 \pm 0.24$ |
| Text (4,096) | S4D | Full model | 0% | 64 | $86.40 \pm 0.21$ |
| | | Uniform $\mathcal{H}_\infty$ | 80% | 13 | $86.02 \pm 0.32$ |
| | | Global $\mathcal{H}_\infty$ | 80% | 13 | $\mathbf{86.20 \pm 0.25}$ |
| | | LAST | 80% | 13 | $85.95 \pm 0.26$ |
| | S5 | Full model | 0% | 192 | $88.88 \pm 0.10$ |
| | | Uniform $\mathcal{H}_\infty$ | 60% | 77 | $82.49 \pm 3.07$ |
| | | Global $\mathcal{H}_\infty$ | 60% | 77 | $\mathbf{88.56 \pm 0.30}$ |
| | | LAST | 60% | 77 | $88.52 \pm 0.20$ |
| Retrieval (4,000) | S4D | Full model | 0% | 64 | $90.46 \pm 0.18$ |
| | | Uniform $\mathcal{H}_\infty$ | 60% | 26 | $\mathbf{89.87 \pm 0.79}$ |
| | | Global $\mathcal{H}_\infty$ | 60% | 26 | $89.84 \pm 0.82$ |
| | | LAST | 60% | 26 | $89.46 \pm 0.58$ |
| | S5 | Full model | 0% | 256 | $91.20 \pm 0.16$ |
| | | Uniform $\mathcal{H}_\infty$ | 50% | 128 | $90.29 \pm 0.30$ |
| | | Global $\mathcal{H}_\infty$ | 50% | 128 | $\mathbf{90.93 \pm 0.34}$ |
| | | LAST | 50% | 128 | $90.42 \pm 0.64$ |
| Image (1,024) | S4D | Full model | 0% | 64 | $77.02 \pm 0.91$ |
| | | Uniform $\mathcal{H}_\infty$ | 0% | 64 | $77.02 \pm 0.91$ |
| | | Global $\mathcal{H}_\infty$ | 0% | 64 | $77.02 \pm 0.91$ |
| | | LAST | 0% | 64 | $77.02 \pm 0.91$ |
| | S5 | Full model | 0% | 256 | $87.30 \pm 0.41$ |
| | | Uniform $\mathcal{H}_\infty$ | 30% | 179 | $86.45 \pm 0.32$ |
| | | Global $\mathcal{H}_\infty$ | 30% | 179 | $\mathbf{87.04 \pm 0.26}$ |
| | | LAST | 30% | 179 | $86.34 \pm 0.37$ |
| Pathfinder (1,024) | S4D | Full model | 0% | 64 | $87.94 \pm 0.70$ |
| | | Uniform $\mathcal{H}_\infty$ | 10% | 58 | $87.59 \pm 0.58$ |
| | | Global $\mathcal{H}_\infty$ | 10% | 58 | $87.20 \pm 0.21$ |
| | | LAST | 10% | 58 | $\mathbf{87.83 \pm 0.66}$ |
| | S5 | Full model | 0% | 256 | $95.15 \pm 0.22$ |
| | | Uniform $\mathcal{H}_\infty$ | 30% | 179 | $71.38 \pm 11.64$ |
| | | Global $\mathcal{H}_\infty$ | 30% | 179 | $57.20 \pm 9.45$ |
| | | LAST | 30% | 179 | $\mathbf{94.45 \pm 0.42}$ |
| Path-X (16,384) | S4D | Full model | 0% | 64 | $88.07 \pm 1.17$ |
| | | Uniform $\mathcal{H}_\infty$ | 10% | 64 | $88.07 \pm 1.17$ |
| | | Global $\mathcal{H}_\infty$ | 10% | 64 | $88.07 \pm 1.17$ |
| | | LAST | 10% | 64 | $88.07 \pm 1.17$ |
| | S5 | Full model | 0% | 256 | $98.41 \pm 0.12$ |
| | | Uniform $\mathcal{H}_\infty$ | 30% | 179 | $90.90 \pm 2.05$ |
| | | Global $\mathcal{H}_\infty$ | 30% | 179 | $69.21 \pm 20.57$ |
| | | LAST | 30% | 179 | $\mathbf{97.95 \pm 0.22}$ |

### E.3 Speech command

Table 10 highlights the results evaluated at the maximum pruning ratio where the accuracy loss of LAST was less than 1%. Figure 8 shows the accuracy at all tested pruning ratios. In Table 10, the state dimension of S4D indicates the average $n_s$ of SISO systems, while in Figure 8, it refers to the average effective state dimension $n$ across layers.

Table 10: Accuracy of pruned models on `Speech Command` task. LAST is evaluated at the maximum tested pruning ratio with less than 1% accuracy loss, and other methods were evaluated for the same pruning ratios.

| Model | Method | Prun. | State dim. | Accuracy (16kHz) | Accuracy (8kHz) |
|-------|--------|-------|-----------|------------------|-----------------|
| S4D | Full model | 0% | 64 | $94.69 \pm 0.10$ | $91.23 \pm 0.93$ |
| | Uniform $\mathcal{H}_\infty$ | 10% | 58 | $94.36 \pm 0.10$ | $90.56 \pm 0.90$ |
| | Global $\mathcal{H}_\infty$ | 10% | 58 | $94.37 \pm 0.09$ | $90.66 \pm 0.99$ |
| | LAST | 10% | 58 | $\mathbf{94.61 \pm 0.07}$ | $\mathbf{90.83 \pm 0.98}$ |
| S5 | Full model | 0% | 128 | $96.43 \pm 0.10$ | $94.26 \pm 0.31$ |
| | Uniform $\mathcal{H}_\infty$ | 20% | 102 | $96.20 \pm 0.11$ | $94.00 \pm 0.18$ |
| | Global $\mathcal{H}_\infty$ | 20% | 102 | $96.21 \pm 0.07$ | $93.91 \pm 0.28$ |
| | LAST | 20% | 102 | $\mathbf{96.31 \pm 0.14}$ | $\mathbf{94.11 \pm 0.36}$ |

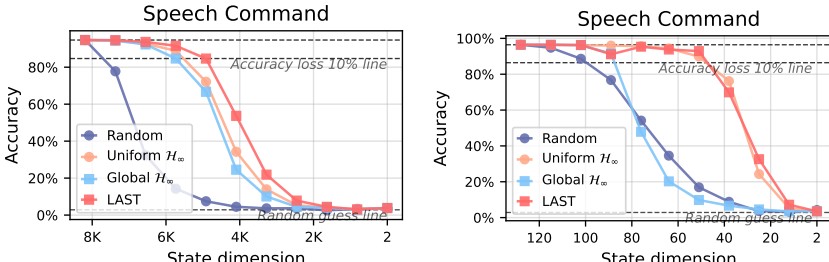

Figure 8: Efficiency-accuracy tradeoff curves of pruned **(Left)** S4D **(Right)** S5 models for `Speech Command` task. LAST obtained more efficient models that maintain performance compared to Uniform $\mathcal{H}_\infty$, which was observed more stably and consistently than Global $\mathcal{H}_\infty$ (LAST w/o score normalization).

