# OpenReview forum: "Layer-Adaptive State Pruning for Deep State Space Models"
_NeurIPS.cc/2024/Conference — NeurIPS 2024 poster_

### Official Review · Reviewer_xVFN · 2024-07-05

**Soundness:** 3
**Presentation:** 2
**Contribution:** 3
**Rating:** 6
**Confidence:** 2

**Summary:**

This submission proposed to conduct pruning on Deep State Space Model (DSSM). Basically, it formulates the output distortion (energy loss) after pruning, the derive that the importance of state is related to $H_{\inf}$ norm, which is used as the pruning criteria. Besides, it proposes to use a greedy optimization by pruning a (entire) subsystem (with the remaining parts intact) to derive the importance of a subsystem (LAST), leading to pruning ratio for this subsystem.

**Strengths:**

1. It is interesting to see pruning in DSSMs and the leading method is coupled with the structure.

**Weaknesses:**

1. Since it is a pioneer work, more comparison method should be used, such as magnitude pruning.

**Questions:**

1. What is the difference between pruning states and parameters?

**Limitations:**

N.A.

---

> ### Author Rebuttal · Authors · 2024-08-07
>
> We thank the reviewer for their feedback and suggestions based on their insightful paper summary. We appreciate their assessment that our method, which is closely coupled with the structure of DSSMs, is particularly interesting.
>
> ### **Weakness:**
> > Since it is a pioneer work, more comparison method should be used, such as magnitude pruning.
>
> We appreciate the reviewer's suggestion to include more comparison methods, such as magnitude pruning. As a common point, this has been thoroughly discussed in Response to Reviewer y832. In summary, we have newly defined magnitude-based state pruning methods and presented the extended experiment results to show clearer effectiveness and contributions. Notably, LAST outperformed other methods by a large margin, and all $\mathcal{H}\_{\infty}$ pruning methods consistently showed superior results over magnitude pruning methods.
>
> ### **Question:**
> > What is the difference between pruning states and parameters?
>
> As in channel pruning (He et al. [2017]), the term of state pruning represents its pruning granularity. That is, the difference arises in the unit of parameters being pruned at once, and the act of pruning training parameters follows the same approach as traditional pruning methods. In a DSSM layer, how the input and output are associated with a state is represented by the training parameters, which include the continuous-time system matrices $\mathbf{\Lambda}$, $\mathbf{B}$, and $\mathbf{C}$, which are discretized into $\overline{\mathbf{\Lambda}}$, $\overline{\mathbf{B}}$, and $\mathbf{C}$. The implementation of state pruning incorporates the identification of insignificant states and masking all parameters corresponding to the states, i.e., masking $\lambda_i$ from $\mathbf{\Lambda}$, the row vector $\mathbf{B}_i$ from $\mathbf{B}$, and the column vector $\mathbf{C}_i$ from $\mathbf{C}$. As compared in Section 4, this concept originates from model reduction studies for single SSMs, and LAST extends this to the deep-layered architecture with nonlinear functions.
>
> To explicitly demonstrate the necessity of state pruning in SSMs, we compared the performance of unstructured random pruning and (structured) random state pruning using the same experimental setup in Response to Reviewer y832. For unstructured random pruning, we pruned randomly selected elements from the system matrices, obtaining the following results:
>
> | Method | Average pruning ratio | Average accuracy loss $\downarrow$ |
> | - | - | - |
> | Unstructured random | 33.00 (36.67) | 59.92 (66.58) |
> | Structured random | 33.00 (36.67) | **29.53 (32.82)** |
>
> Despite the same amount of parameters being removed, it can be observed that the model completely loses its performance when pruning is performed in an unstructured manner.
>
> This is because unstructured pruning can disrupt the integrity of the model's learned dynamics by altering subsystems into significantly different ones. In contrast, state pruning maintains the functionality of subsystems, leading to less performance degradation. This highlights the importance of considering the structure and relationships within the model when applying pruning techniques.
>
> ---
>
> Y. He et al. Channel pruning for accelerating very deep neural networks. In *The IEEE International Conference on Computer Vision*, 2017.

---

> > ### Comment · Reviewer_xVFN · 2024-08-13
> > **Response to Author**
> >
> > Thanks for your detailed response and appended experiments.
> >
> > For my first concern on magnitude pruning, author has defined a similar experiment setting for DSSM. Though author claimed that the parameters pruning and DSSM pruning is quite different in rebuttal to y832, I did not catch the main idea of the difference.
> >
> > Besides, author further claimed that the `there is a fundamental disparity between unstructured and state pruning methods in SSMs, making it unfair to compare them` in discussion with xVFN, which is me. I fail to see the disparity explanation, is that the anwser to my question?
> >
> > Similar situation happens in the anwser to the question, I fail to understand the explanation. Overall, I keep my score for I can not fully understand DSSM and how it is different with normal deep nerual network, especially in pruning.

---

> > > ### Author Response · Authors · 2024-08-14
> > >
> > > Thank you for taking the time to read our rebuttal. Due to space constraints, we had to distribute our responses across several rebuttals. We appreciate your thorough review of all the details.
> > >
> > > ### **Correction of misunderstanding**
> > >
> > > We apologize for any confusion caused by our use of 'disparity' and for not clearly referencing the relevant response to the reviewer's question.
> > >
> > > To clarify, we used 'disparity' to describe the performance gap between unstructured pruning and state pruning (a type of structured pruning), which is evident in our response.
> > >
> > > As shown in the previous rebuttal, structured pruning (state pruning) outperforms unstructured pruning even in randomly pruned cases, which clearly shows the effect of using state pruning granularity by excluding the other factors that might affect the performance. Thus, what we intended to deliver to Reviewer y832 was that **the presented table contains only state pruning**, omitting unstructured pruning due to the significant performance difference between unstructured/structured versions of every pruning method.
> > >
> > > In another way, our intention may be better conveyed by replacing 'fundamental' with 'trivial.'
> > >
> > > Therefore, that part does not conflict with the explanation that state pruning is a type of neural network pruning but is tailored for SSM.
> > >
> > > ### **Additional comments to the previous rebuttal**
> > >
> > > The equations used in our explanation are based on the standard form of system matrices in SSM, consistent with the notation in our submission. To assist the reviewer's understanding, we define key terminology and a detailed illustration of state pruning.
> > >
> > > Pruning granularity refers to the specific pattern or structure of parameters to be pruned, such as individual weights, entire neurons, channels, or states. For example, channel pruning prunes all parameters corresponding to an insignificant channel. Likewise, state pruning prunes all parameters corresponding to an insignificant state.
> > >
> > > For a DSSM layer's training parameters $\mathbf{\Lambda}$, $\mathbf{B}$, and $\mathbf{C}$, such that $x_{k+1}=\mathbf{\Lambda}x_k+\mathbf{B}u_k$ and $y_k=\mathbf{C}x_k$, the specific parameters for a state $i$ are the $i$th diagonal element $\lambda_i$ of $\mathbf{\Lambda}$, the $i$th row vector $\mathbf{B}_i$ of $\mathbf{B}$, and the $i$th column vector $\mathbf{C}_i$ of $\mathbf{C}$. For instance, if the 2nd state is identified as insignificant in a DSSM layer with the state dimension of 4, by masking the insignificant parameters to 0, the system matrices before and after pruning would be as follows:
> > >
> > > $x_{k+1}=\begin{bmatrix} \*&&& \\\ &0&& \\\ &&\*& \\\ &&&\*\end{bmatrix}x_k + \begin{bmatrix} \*&\*&\*&\* \\\ 0&0&0&0 \\\ \*&\*&\*&\* \\\ \*&\*&\*&\* \end{bmatrix} u_k$
> > >
> > > $y_{k}=\begin{bmatrix} \*&0&\*&\* \\\ \*&0&\*&\* \\\ \*&0&\*&\* \\\ \*&0&\*&\* \end{bmatrix}x_k$
> > >
> > > In this sense, we would like to reiterate that we responded to the question 'What is the difference between pruning states and parameters?' by 'As in channel pruning (He et al. [2017]), the term of state pruning represents its pruning granularity. That is, the difference arises in the unit of parameters being pruned at once, and the act of pruning training parameters follows the same approach as traditional pruning methods'.

---

> > > ### Author Response · Authors · 2024-08-14
> > > **Toy example of the proposed method**
> > >
> > > We understand that DSSM might be relatively new to some reviewers, having been proposed only a few years ago and primarily applied to sequential data. However, it is a well-established attention-drawing network rooted in linear system theory, demonstrating superior performance on sequences with continuous properties. Similar to how other works have developed structured pruning for specific models like the diffusion model (G. Fang et al. [2023]), our work offers a network optimization method tailored to DSSMs.
> > >
> > > To illustrate DSSM and its pruning, we present an explicit time-domain toy example evaluating state importance. Consider a simple single-input single-output (SISO) system with a four-dimensional state vector parameterized and trained in a DSSM layer. The numerical values in the matrices represent the layer’s training parameters, totaling 12 in this example.
> > >
> > > **Example system of a DSSM layer**
> > >
> > > $x_{k+1}=\begin{bmatrix}0.9&&& \\\ &0.7&& \\\ &&0.5& \\\ &&&0.3\end{bmatrix}x_k+\begin{bmatrix}1 \\\ 1 \\\ 1 \\\ 1 \end{bmatrix}u_k$
> > >
> > > $y_{k}=\begin{bmatrix}1&1&1&1\end{bmatrix}x_k$
> > >
> > > Assuming the input consistently comes in as 1 and $x_{-1}=[0,0,0,0]$, the state and output of the given system would be as follows:
> > >
> > > | | Input $u_k$ | State $x_k$ | Output $y_k$ |
> > > | - | - | - | - |
> > > | $k=0$ | $1$ | $[1,1,1,1]$ | $4$ |
> > > | $k=1$ | $1$ | $[1.9,1.7,1.5,1.3]$ | $6.4$ |
> > > | $k=2$ | $1$ | $[2.71,2.19,1.75,1.39]$ | $8.04$ |
> > >
> > > As the state-output matrix is $[1,1,1,1]$, the state directly contributes to the output. Although the diagonal terms $\lambda_i$ are linear, e.g., 0.9, 0.7, 0.5, 0.3, their contributions to the output show nonlinear differences, e.g., 2.71, 2.19, 1.75, 1.39. This nonlinearity occurs because each state component, influenced by the diagonal term, decays at different rates, leading to varying impacts on the output.
> > >
> > > Thus, we learn that the magnitude of $\lambda_i$ alone cannot fully determine the relative importance of each state. When the $B$ and $C$ matrices vary from this simple example, their effects must also be considered. Moreover, since DSSM processes sequence data, not one-shot data, the analysis of how the model processes data can be more intuitive and comprehensive in the frequency domain. As explained in the Response to Reviewer y832, the $\mathcal{H}\_{\infty}$ norm measures the maximum frequency-domain gain of each subsystem, and the proposed method even considers the contribution of each layer to the entire model.
> > >
> > > Specifically, $\mathcal{H}\_{\infty}$ and LAST scores can evaluate the importance of each state (subsystem) in the previous example system, as follows:
> > >
> > > | State | $\mathcal{H}\_{\infty}$ score | LAST score |
> > > | - | - | - |
> > > | $\lambda_0$ | $(1\cdot 1)/(1-0.9)^2=100$ | $100/100=1$ |
> > > | $\lambda_1$ | $(1\cdot 1)/(1-0.7)^2=11.11$ | $11.11/(11.11+100)=0.099$ |
> > > | $\lambda_2$ | $(1\cdot 1)/(1-0.5)^2=4$ | $4/(4+11.11+100)=0.034$ |
> > > | $\lambda_3$ | $(1\cdot 1)/(1-0.3)^2=2.04$ | $2.04/(2.04+4+11.11+100)=0.017$ |
> > >
> > > For both methods, insignificant states can be identified by their smaller scores, allowing us to prune the corresponding parameters. Although not explicitly demonstrated in this example, the LAST score is particularly useful when dealing with multiple DSSM layers, as it enables the comparison of states across different layers by evaluating the relative contribution of each subsystem to the entire model.
> > >
> > > Consequently, by pruning the two insignificant states (i.e., pruning 6 corresponding parameters), the resulting pruned DSSM reduces the total number of parameters by 6, as shown below.
> > >
> > > $x_{k+1}=\begin{bmatrix}0.9& \\\ &0.7\end{bmatrix}x_k+\begin{bmatrix}1 \\\ 1\end{bmatrix}u_k$
> > > $y_{k}=\begin{bmatrix}1&1\end{bmatrix}x_k$
> > >
> > > ---
> > > G. Fang et al. Structural pruning for diffusion models. In *Advances in Neural Information Processing Systems*, 2023.

---

### Official Review · Reviewer_y832 · 2024-07-08

**Soundness:** 3
**Presentation:** 3
**Contribution:** 2
**Rating:** 5
**Confidence:** 2

**Summary:**

In this paper, the authors propose Layer-Adaptive $\mathcal{H}{\infty}$ STate pruning (LAST), a deep state space model (SSM) pruning method that optimizes the state dimension of a deep diagonal SSM in terms of model-level energy loss. Experimental results on different tasks demonstrate that the proposed method performs better than Uniform $\mathcal{H}{\infty}$ and Global $\mathcal{H}{\infty}$ methods.

**Strengths:**

The paper proposes a novel method to prune deep state space models. Experimental results on different tasks demonstrate the effectiveness of the proposed methods.

**Weaknesses:**

In the experiments, the work only compares the proposed method with Uniform $\mathcal{H}{\infty}$ and Global $\mathcal{H}{\infty}$ methods. It would be good if more weight pruning methods could be applied to deep state space models for fair comparisons.

**Questions:**

Could the structured weight pruning methods on the traditional DNNs be applied to the pruning of deep state space models?

**Limitations:**

The authors have adequately addressed the limitations.

---

> ### Author Rebuttal · Authors · 2024-08-07
>
> We are glad the reviewer appreciated the novelty and effectiveness of the proposed method. As the reviewer suggested, we compare our method to three magnitude pruning methods that use magnitude-based criteria from traditional DNNs and state pruning granularity as in the proposed method.
>
> State pruning is structured pruning, where all training parameters corresponding to the insignificant states are pruned based on a state importance metric. In traditional DNN, such as MLP, the transfer function of a layer is a weight matrix, $f(u)=Wu$, where each element $W_{ij}$ is a weight parameter. This allows evaluating the training parameters by their magnitude, $|W_{ij}|$, as the magnitude controls the input's influence on the output. Applying this to SSMs could involve unstructured magnitude pruning, which masks the high-magnitude elements of each system matrix according to the pruning ratio.
>
> However, as discussed in Response to Reviewer xVFN, there is a fundamental disparity between unstructured and state pruning methods in SSMs, making it unfair to compare them. Thus, we define magnitude-based state pruning methods for SSMs and evaluate their **ability to distinguish significant and insignificant states**, thereby preserving overall model performance.
>
> - **Uniform Magnitude**
>
>     Every layer is pruned to have an identical pruning ratio by evaluating the importance of each state $i$ as $|\overline{\lambda}_i|||\mathbf{\overline{B}}_i||||\mathbf{C}_i||$. The pruning ratio is applied uniformly across all layers. While any $L_p$ norm can be used, we present the results using the $L_2$ norm as an example.
>
> - **Global Magnitude**
>
>     The same state importance metric as in Uniform Magnitude is used, but the comparison group is extended from intra-layer to inter-layer, ensuring the pruning ratio is met globally across the entire network.
>
> - **Layer-Adaptive Magnitude-based Pruning (LAMP)**
>
>     This method employs a metric of $\frac{|\overline{\lambda}_i|^2||\mathbf{\overline{B}}_i||^2||\mathbf{C}_i||^2}{\sum\_{j\geq i}|\overline{\lambda}_j|^2||\mathbf{\overline{B}}_j||^2||\mathbf{C}_j||^2}$ adapted from Lee et al. [2021], which originally used $\frac{W_i^2}{\sum\_{j\geq i} W_j^2}$ for a real-valued weight parameter $W$. The state indices in the denominator are assumed to be ordered based on their evaluation using the basic magnitude criteria.
>
> Extending Table 1, we report newly conducted JAX experiments with 3 different seeds on an NVIDIA RTX 3090 as follows:
>
> | Method | Average pruning ratio | Average accuracy loss $\downarrow$ |  State importance |
> | - | - | - | - |
> | Random | 33.00 (36.67) | 29.53 (32.82) | - |
> | Uniform magnitude | 33.00 (36.67) | 22.03 (24.48) | $\vert\overline{\lambda}_i\vert\Vert\mathbf{\overline{B}}_i\Vert\Vert\mathbf{C}_i\Vert$ |
> | Global magnitude | 33.00 (36.67) | 17.49 (19.43) | $\vert\overline{\lambda}_i\vert\Vert\mathbf{\overline{B}}_i\Vert\Vert\mathbf{C}_i\Vert$ |
> | LAMP | 33.00 (36.67) | 18.07 (20.07) | $\frac{\vert\overline{\lambda}_i\vert^2\Vert\mathbf{\overline{B}}_i\Vert^2\Vert\mathbf{C}_i\Vert^2}{\sum\_{j\geq i}\vert\overline{\lambda}_j\vert^2\Vert\mathbf{\overline{B}}_j\Vert^2\Vert\mathbf{C}_j\Vert^2}$ |
> | Uniform $\mathcal{H}\_{\infty}$ | 33.00 (36.67) | 4.32 (4.80) | $\frac{\Vert\mathbf{C}_i\Vert^2\Vert\mathbf{\overline{B}}_i\Vert^2}{(1-\vert\overline{\lambda}_i\vert)^2}$ |
> | Global $\mathcal{H}\_{\infty}$ | 33.00 (36.67) | 7.51 (8.35) | $\frac{\Vert\mathbf{C}_i\Vert^2\Vert\mathbf{\overline{B}}_i\Vert^2}{(1-\vert\overline{\lambda}_i\vert)^2}$ |
> | LAST | 33.00 (36.67) | **0.52 (0.58)** | $\frac{\frac{\Vert\mathbf{C}_i\Vert^2\Vert\mathbf{\overline{B}}_i\Vert^2}{(1-\vert\overline{\lambda}_i\vert)^2}}{\sum\_{j\geq i}\frac{\Vert\mathbf{C}_j\Vert^2\Vert\mathbf{\overline{B}}_j\Vert^2}{(1-\vert\overline{\lambda}_j\vert)^2}}$ |
>
> The table lists the accuracy loss of pruning methods for S5 models (Smith et al. [2023]), averaged across all 10 tasks. Values in parentheses exclude non-compressible tasks with zero pruning ratios and accuracy loss.
>
> At the same pruning ratio, LAST significantly outperforms other methods by exhibiting the least accuracy loss.
>
> Random state pruning results show that all methods can identify insignificant states, but all $\mathcal{H}\_{\infty}$ pruning methods surpass magnitude pruning methods. This implies the suitability of $\mathcal{H}\_{\infty}$ pruning for SSMs, which can be explained by the transfer functions of SSMs. In an SSM layer, the multiplicative mapping from input to output in the frequency domain is given by $\mathbf{Y}(z)=\mathbf{C}(zI-\mathbf{\overline{\Lambda}})^{-1}\mathbf{\overline{B}}\mathbf{U}(z)$, where $\mathbf{U}$ and $\mathbf{Y}$ are the $\mathcal{Z}$-transforms of input and output sequences $\mathbf{u}$ and $\mathbf{y}$, and $\mathbf{C}(zI-\mathbf{\overline{\Lambda}})^{-1}\mathbf{\overline{B}}$ is the transfer function matrix.
>
> The importance of training parameters in SSMs can be evaluated by their influence in the frequency domain. Using the $\mathcal{H}\_{\infty}$ norm, which measures the maximum gain of the transfer function in the frequency domain, $\mathcal{H}\_{\infty}$ pruning optimizes the system by **minimizing the maximum gain alteration**, leading to better performance than magnitude pruning. Specifically, $\overline{\lambda}_i$'s importance in $\overline{\mathbf{\Lambda}}$ is measured by $(1-|\overline{\lambda}_i|)^{-1}$ in $\mathcal{H}\_{\infty}$ pruning, while magnitude pruning measures $|\overline{\lambda}_i|$. Additionally, LAST normalizes this importance by the total energy transmission of the layer, preventing extensive pruning of high energy-conveying layers and enabling stable global comparison and pruning.
>
> ---
>
> J. T. Smith et al. Simplified state space layers for sequence modeling. In *The International Conference on Learning Representations*, 2023.
>
> J. Lee et al. Layer-adaptive sparsity for the magnitude-based pruning. In *The International Conference on Learning Representations*, 2021.

---

> > ### Comment · Reviewer_y832 · 2024-08-13
> >
> > Thank you for the detailed responses and extra experimental comparisons. However, the comparisons with magnitude-based weight pruning methods cannot fully address my concern as magnitude-based pruning is a very straightforward approach. I highly suggest that the authors compare the proposed method with more well-designed approaches. In conclusion, I would not oppose to accept this paper. I would like to keep my rating as 5: Borderline accept.

---

> > > ### Author Response · Authors · 2024-08-14
> > >
> > > We appreciate the reviewer's careful review and further feedback on our rebuttal. We agree that comparing the proposed method with other pruning methods beyond magnitude-based pruning could provide a clearer validation of our work.
> > >
> > > While magnitude-based pruning is straightforward, it is an effective and strong baseline, as it continues to be widely used in studies on iterative pruning, pruning schedules, or pruning on large language models (Sun et al. [2024], K. Xu et al. [2023], M. Gupta [2024]). Notably, LAST and $\mathcal{H}\_{\infty}$ pruning significantly outperform it, highlighting its necessity in SSM. Additionally, the previous comparison can be viewed as one for one-shot pruning methods, which are particularly useful when dealing with scaled models compared to iterative methods.
> > >
> > > In response to the remaining reviewer's concern, we present the following results for a scaling-based pruning method using batch normalization statistics as another baseline.
> > >
> > > **Uniform Scaling-based Pruning** (Liu et al. [2017])
> > >
> > > This method uses the scale $\gamma^{(l)}\in\mathbb{R}^h$ trained in the $l$th batch normalization layer as the criterion for channel pruning out of $h$ channels. Specifically, we used $\gamma^{(l-1)}$ to prune $\mathbf{B}^{(l)}$ and $\gamma^{(l)}$ to prune $\mathbf{C}^{(l)}$ in consideration of the input-output processing scheme of SSM.
> > >
> > > | Method | Average pruning ratio | Average accuracy loss $\downarrow$ | Importance |
> > > | - | - | - | - |
> > > | Uniform scaling-based | 33.00 (36.67) | 25.84 (28.71) | $\gamma_k^{(l-1)}$ and $\gamma_k^{(l)}$ for channel $k$ |
> > >
> > > Due to time constraints, we could not test pruning methods that involve retraining, such as using a designed loss function or tracking gradient flow. We promise to include comparison experiments with other well-designed pruning methods in the final version of the paper.
> > >
> > > Additionally, although we considered recent feature statistics-based pruning methods, we noticed that further research is needed, especially in observing specific patterns in feature vectors within SSM. As we are the first to explore pruning in SSM, designing new criteria for state pruning is akin to suggesting a new algorithm. We hope the reviewer considers our work a foundational study that suggests appropriate pruning criteria and directions for SSM.
> > >
> > > ---
> > >
> > > M. Sun et al. A simple and effective pruning approach for large language models. In *The International Conference on Learning Representations*, 2024.
> > >
> > > K. Xu et al. Efficient joint optimization of layer-adaptive weight pruning in deep neural networks. In *Proceedings of the IEEE/CVF International Conference on Computer Vision*, 2023.
> > >
> > > M. Gupta, et al. Is complexity required for neural network pruning? a case study on global magnitude pruning. In IEEE Conference on Artificial Intelligence, 2024.
> > >
> > > Z. Liu et al. Learning efficient convolutional networks through network slimming. In *Proceedings of the IEEE international conference on computer vision*, 2017.

---

### Official Review · Reviewer_c1Mv · 2024-07-17

**Soundness:** 3
**Presentation:** 3
**Contribution:** 3
**Rating:** 6
**Confidence:** 3

**Summary:**

Inspired by the traditional layer-adaptive neural network pruning, this paper develops and verifies a layer-adaptive model order reduction (MOR) method to reduce the state dimension in DSSM models. The proposed method reveals state importance and prunes insignificant subsystems for a desired compression level, reducing unnecessary computations while bounding the output energy loss.

**Strengths:**

1. Proposes to use layer-adaptive pruning methods in new areas DSSM models.

2. Extensive experiments among 10 tasks show the robustness and effectiveness of the methods. Besides, it conducts ablation studies on hyper-parameters in Appendix.

**Weaknesses:**

Since I am not an expert in DSSMs area, I just doubt the impact of the state dimension to the computational cost, it might be better to provide some statistics to show the impact. Besides, if this state dimension is very significant, do we have other some related works but not pruning to reduce the state dimension? Can you elaborate more on related works on reducing state dimension?

**Questions:**

As mentioned in the weakness,

1) please use some statistics to show the importance of state dimension?

2) if the state dimension is significant, can you provide more related works on reducing the state dimension, maybe those related works are not from the perspective of pruning to reduce the state dimension, but i think it is meaningful to discuss them.

---

> ### Author Rebuttal · Authors · 2024-08-07
>
> We appreciate the reviewer's assessment that the paper contributes to new areas of DSSMs and the experiments are thorough to show the robustness and effectiveness of the proposed method. The reviewer's concerns were the practical impacts and related works of reducing state dimension.
>
> > Please use some statistics to show the importance of state dimension.
>
> We present the average evaluation step speed and peak GPU memory usage when the state dimension is reduced to the maximum pruning ratio that resulted in less than 1% accuracy loss using our proposed method. The evaluation was conducted by implementing an actual decrease in model size on an NVIDIA RTX 3090. While there is some variance depending on the channel size of each model per task, reducing the state dimension improves efficiency in both computational and memory costs.
>
> |  | ListOps | Text | Retrieval | Image | Pathfinder | Path-X |
> | - | - | - | - | - | - | - |
> | Pruning ratio | 0% | 60% | 50% | 30% | 30% | 30% |
> | Inference speed $\uparrow$ | 1.0$\times$ | 1.6$\times$ | 1.7$\times$ | 1.2$\times$ | 1.1$\times$ | 1.3$\times$ |
> | GPU memory usage $\downarrow$ | 1.0$\times$ | 0.9$\times$ | 0.6$\times$ | 1.0$\times$ | 0.8$\times$ | 0.8$\times$ |
>
> > If the state dimension is significant, can you provide more related works on reducing the state dimension, maybe those related works are not from the perspective of pruning to reduce the state dimension, but I think it is meaningful to discuss them.
>
> To alleviate the issue of large state dimensions, related research has evolved with two primary approaches: architectural and algorithmic.
>
> **1. Architectural approaches**
>
> One architectural approach is modulating states for parallelized inference. Specifically, an $h\cdot n_{SISO}$-th order system is modulated by $h$ systems, each with $n_{SISO}$ states, where $h$ is the number of channels. This means each $n_{SISO}$-th order system handles information from a specific channel. To address the inability to integrate information across channels, a channel mixing layer is introduced. This is referred to as, in our paper, the multi-SISO architecture (Gu et al. [2021], Gu et al. [2022a]), allowing for training $h\cdot n_{SISO}$ effective state spaces with reduced computational burden by parallelizing into $n_{SISO}$, thus enabling the development of many subsequent SSMs (Gupta et al. [2022], Gu et al. [2022b]).
>
> In comparison, another architecture proposed by Smith et al. [2023], referred to as MIMO architecture, uses $n_{MIMO}\ll h\cdot n_{SISO}$ states to determine the output for a channel based on information from all input channels, making the mixing layer unnecessary.  This parameterization allows for high performance with much smaller state dimensions than equivalent block systems in multi-SISO layers. For instance, in the Path-X task, which involves the longest sequences among LRA tasks, this model shows state-of-the-art performance. However, both architectures lack optimization methods for model size, leading to computational inefficiencies when the task does not require such a high model capacity.
>
> - **Relation to the pruning approach**
>
>     Our state pruning method can be applied to all these architectures, effectively reducing the model size to fit task or resource requirements while preserving the performance. It identifies and adaptively prunes the states with the least impact on performance based on layer-wise energy transmission. In multi-SISO architectures, since each state is assigned to a specific channel, the difference in importance among states may not be significant. However, our method still shows superior retention compared to other methods. In MIMO architectures, the difference in state importance becomes more pronounced, and the accuracy loss caused by pruning varies significantly depending on how a method evaluates the state importance. Consequently, our method effectively preserves the most significant states and demonstrates superior performance compared to other pruning methods, as shown in the table in Response to Reviewer y832.
>
> **2. Algorithmic approaches**
>
> One recent study has shifted from learning the element of decomposition of the transfer function in DSSMs to directly learning the coefficients of the transfer function (Parnichkun et al. [2024]). The motivation behind this approach is also to address the computational load caused by large state dimensions. The primary contribution of this approach is making computations even independent of the state dimension. However, as discussed in Appendix A, this approach has a weakness of only guaranteeing stability at initialization, unlike DSSMs, which can consistently guarantee stability due to their direct parameterization for the poles of the transfer function. Therefore, as a branch of DSSM research, our work contributes to enhancing the efficiency of DSSMs, which is dependent on the state dimension, by minimizing the state dimension.
>
> ---
>
> A. Gu et al. Combining recurrent, convolutional, and continuous-time models with linear state space layers. In *Advances in neural information processing systems*, 2021.
>
> A. Gu et al. Efficiently modeling long sequences with structured state spaces. In *The International Conference on Learning Representations*, 2022a.
>
> A. Gupta et al. Diagonal state spaces are as effective as structured state spaces. In *Advances in Neural Information Processing Systems*, 2022.
>
> A. Gu et al. On the parameterization and initialization of diagonal state space models. In *Advances in Neural Information Processing Systems*, 2022b.
>
> J. T. Smith et al. Simplified state space layers for sequence modeling. In *The International Conference on Learning Representations*, 2023.
>
> R. N. Parnichkun et al. State-free inference of state-space models: The transfer function approach, In *International Conference on Machine Learning*, 2024.

---

> > ### Comment · Reviewer_c1Mv · 2024-08-13
> >
> > Thank you for the authors' response and address my concerns. I would like to keep my positive score.

---

### Author Rebuttal · Authors · 2024-08-07

We thank all reviewers and ACs for their efforts in reviewing our paper. We are glad that the reviewers found the proposed method novel and noted that the experiments support its effectiveness. We sincerely appreciate the insightful and constructive feedback, and we have carefully responded to all comments with individual replies.

**Highlights of rebuttal**
- Clearer descriptions of the motivation to reduce state dimension, with comparisons to related works
- Deeper explanations for why the proposed state importance metric works well for state space models
- Extended experiments with magnitude pruning
- Detailed comparison between unstructured pruning and state pruning
- Demonstrations showing the computational cost reduction by LAST

---

### Author Response · Authors · 2024-08-12
**LaTex Rendering Issue**

Dear Reviewers and Area Chairs,

Thank you for taking the time to review our submission.

We would like to bring to your attention a concern regarding the LaTeX rendering on the OpenReview platform. Due to a rendering error, some LaTeX formulas in the Title, Abstract, and Rebuttals may not display correctly on the page. This issue can usually be resolved by refreshing the page.

However, we have noticed that the Abstract does not always render correctly, even after refreshing. We kindly ask you to refer to the Abstract in the submission for accurate information.

We look forward to engaging in the discussion process and hope this does not cause any misunderstandings during the remainder of the discussion period.

Sincerely,

Authors.

---

### Decision · Program_Chairs · 2024-09-25

**Decision:**

Accept (poster)

**Comment:**

Thanks for your submission to NeuIPS.

This paper was borderline, leaning to accept, prior to the rebuttal and discussion period.  There were questions/concerns about the state dimension, along with requests for some additional experiments.  The authors provided a very thorough and useful rebuttal, addressing these issues satisfactorily.  In general, the reviewers agreed (with possibly some lingering questions still at the end), but ultimately all three reviewers indicated that they were OK accepting the paper.  As a result I am happy to recommend acceptance of this paper.

Please do add the additional results from the rebuttal into the final manuscript.